# Estimating Time Series Foundation Model Transferability via In-Context Learning

## Abstract

Time series foundation models (TSFMs) offer strong zero-shot forecasting via large-scale pre-training, yet fine-tuning remains critical for boosting performance in domains with limited public data. With the growing number of TSFMs, efficiently identifying the best model for downstream fine-tuning becomes increasingly challenging. In this work, we introduce TimeTic, a transferability estimation framework that recasts model selection as an in-context-learning problem: given observations on known (source) datasets, it predicts how a TSFM will perform after fine-tuning on a downstream (target) dataset. TimeTic flexibly organizes observed model–data relationships as contextual information, allowing it to adapt seamlessly to diverse test-time scenarios. Leveraging the natural tabular structure formed by dataset meta-features, model characteristics, and fine-tuned performance, we employ tabular foundation models to serve as in-context learners. We further introduce a novel model characterization based on entropy evolution across model layers, capturing embedding-space distinctions and enabling TimeTic to generalize across arbitrary model sets. We establish a comprehensive benchmark for transferability estimation including 10 datasets, 10 foundation models, and 3 forecasting tasks. On this benchmark, TimeTic's estimation demonstrates strong alignment with actual fine-tuned performance for previously unseen datasets, achieving a mean rank correlation of approximately 0.6 and a 30% improvement compared to using zero-shot performance as the transferability score. Source code is available at `https://anonymous.4open.science/r/ICLR2026-TimeTic-3975`.

## 1 Introduction

The emergence of time series foundation models (TSFMs) is reshaping the paradigm of time series forecasting (Liang et al., 2025) through their strong zero-shot capabilities. Although efficient and cost-effective, zero-shot inference often underperforms in out-of-distribution scenarios, particularly in domains with limited public data, such as healthcare (Gupta et al., 2024) and finance (Fu et al., 2024). Fine-tuning helps bridge the gap by transferring generalized knowledge from large-scale pre-training to specific, resource-limited downstream tasks (Li & Zhu, 2025). However, due to the inherent diversity of time series data, no single model consistently outperforms others in all scenarios (Brigato et al., 2025). Selecting the most appropriate model from all available models becomes a critical consideration that directly impacts the performance of downstream tasks (Ding et al., 2024). A straightforward approach would be to enumerate all available TSFMs and evaluate their fine-tuned performance, but this is impractical due to the significant computational cost and extensive training time required, as shown in Figure 1 (a). Therefore, a crucial question arises: *how can we efficiently identify the best candidate time series model to fine-tune for a given test-time scenario with limited data?*

Existing efficient model selection techniques generally fall into two categories: (1) statistical metrics (You et al., 2021; Nguyen et al., 2023) and (2) meta-learning strategies (Öztürk et al., 2022; Abdallah et al., 2022b). Most statistical metrics are designed for image classification and depend on strong assumptions about the class structure (Li et al., 2021; Gholami et al., 2023). Although computationally efficient, they are predefined and uniformly applied across scenarios, limiting their adaptability to diverse time series forecasting tasks and models. Meta-learning methods instead train

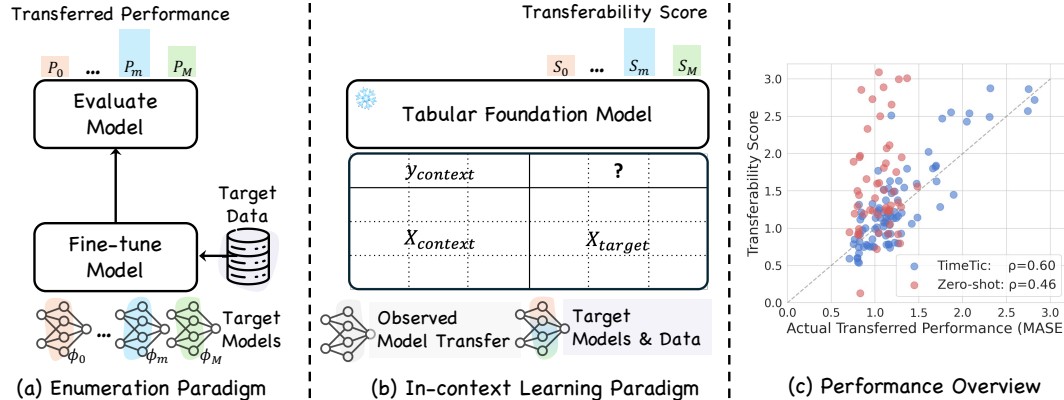

Figure 1: Model selection paradigms. **(a)** Enumeration paradigm: Each TSFM is fine-tuned on the target data, and their performances $P_m$ are evaluated to select the best model. **(b)** In-context learning paradigm: Observed model transfers are organized into a context table ($X_{\text{context}}, y_{\text{context}}$) composed of characteristic–performance pairs. This table provides exemplars for a tabular foundation model, which then predicts the transferred performance $S_m$ of a target model on new data, given its target table $X_{\text{target}}$. **(c)** Performance overview: The transferability scores estimated by TIMETIC show a strong alignment with actual fine-tuned performance, achieving more than a 30% higher Spearman rank correlation compared to ranking models based on their zero-shot performance.

a meta-estimator on task-performance pairs to predict fine-tuned performance. However, the estimator is tied to its (fixed) training corpus and a predefined model set, restricting its ability to generalize to new tasks or models. In general, existing approaches lack the adaptability needed for transferability estimation in practical settings with TSFMs, where test-time scenarios are open-ended and constantly evolving.

In this study, we present TIMETIC, a framework for estimating the transferability of TSFMs by casting performance prediction as an in-context learning task: given a model's transferred performance on known datasets, predict its finetuned performance on a new target dataset. As illustrated in Figure 1(b), this paradigm allows flexible organization of historical data to make informed predictions. To this end, we integrate past observations into a tabular representation, consolidating models, datasets, and transferred performance within a structured table. This format not only facilitates scalability with growing observational data but also clearly captures interrelationships among entities. Recent advances in tabular foundation models have demonstrated strong in-context learning capabilities for structured data (Robertson et al., 2025; Hollmann et al., 2025). Building on this, we employ a tabular foundation model as the in-context learner, enabling efficient prediction of target model performance from past transfer observations. To scale across a growing variety of TSFMs, we further introduce a novel model characterization strategy based on entropy evolution across layers. This architecture-agnostic approach allows TIMETIC to generalize effectively to various types of models. Extensive experiments on 10 datasets, 10 TSFMs, and 3 forecast settings demonstrate that TIMETIC consistently outperforms existing methods, achieving an average Spearman rank correlation of approximately 0.6 and delivering a 30% improvement over rankings based on zero-shot performance, as shown in Figure 1(c).

The main contributions of this paper are summarized as follows:

- We propose TIMETIC, the first in-context transferability estimation framework for TSFMs, leveraging tabular foundation models to predict fine-tuned performance from an arbitrary number of past transfer observations. This offers a more practical and efficient alternative to current methods.

- We introduce a model-agnostic characterization of TSFMs based on the entropy profile, the trajectory of token sequence entropy across model layers. This enables TIMETIC to estimate transferability on unseen model classes, without being restricted to a fixed candidate set.

- We construct a comprehensive transferability benchmark that spans 10 widely used datasets, 10 time series foundation models, and 3 forecasting tasks, and demonstrate that TIMETIC outperforms existing approaches by more than 30% in model transferability estimation.

## 2 RELATED WORK

**Time series foundation model** Time series forecasting is critical to decision making, driving advances in both statistical and domain-specific deep learning approaches (Liang et al., 2024). Recently, the focus has shifted to TSFMs because of their strong generalization. Transformer has become the dominant architecture in TSFMs, which fall into three categories: (1) *Encoder-only models*, such as Moirai (Woo et al., 2024) and Moment (Goswami et al., 2024), using mask prediction for forecasting. (2) *Encoder-decoder models*, exemplified by the Chronos family (Ansari et al., 2024), which adapts T5 (Raffel et al., 2019) with quantization-based tokenization for time series forecasting. (3) *Decoder-only models*, including TimesFM (Das et al., 2023), Lag-Llama (Rasul et al., 2023), Timer (Liu et al., 2024) and Time-MoE (Shi et al., 2025), employing autoregressive generation for future prediction.

**Transferability metric** Assessing the transferability of pretrained models is essential for model selection (Okanovic et al., 2024; Lin et al., 2024). Transferability metrics generally aim to quantify the statistical relationship between feature embeddings and sample labels. Most metrics such as H-Score (Bao et al., 2019a), NCE (Tran et al., 2019) and LEEP (Nguyen et al., 2020) are primarily designed for classification tasks, relying on the assumption that model outputs follow a categorical distribution. However, in most TSFMs, the final output is continuous, making these metrics nonsensical without discretization. Only a few metrics such as LFC (Deshpande et al., 2021), LogME (You et al., 2021), and RegScore (Nguyen et al., 2023) are applicable in broader tasks by estimating transferability through similarity of the characteristic of the label, marginal likelihood and linear regression error, respectively.

**Learning to select** Early work (Lemke & Gabrys, 2010) explored meta-learning strategies that leverage time series characteristics to predict the performance of forecasting models, demonstrating that model accuracy often correlates with data properties. Along this line, FFORMPP (Talagala et al., 2019) and AutoForecast (Abdallah et al., 2022a) train meta-estimators - Bayesian and mixed architecture, respectively - on feature-performance pairs to identify the best model from a predefined pool. Instead of feature-based regression, SeqFusion (Huang et al., 2025) embeds both time series and candidate models into a shared representation space, allowing selection via similarity search. However, its effectiveness heavily depends on encoder quality (Zhang et al., 2023; Meng et al., 2023), which is difficult to guarantee for unseen models or data. More recently, Wei et al. (2025) have probed LLMs for model selection by encoding the model and data information in prompts and relying on LLM reasoning. Although promising, such approaches remain unreliable due to their opacity. In general, despite progress, generalizable model selection, scalable to unseen models and datasets, remains an open challenge. In particular, with the rapid proliferation of TSFMs, model selection method for TSFMs is still unexplored.

## 3 METHODOLOGY

**Problem setup** In model selection, we consider a set of $M$ candidate models $\{\phi_i\}_{i=1}^M$, and a target dataset $D$. Each model has a ground truth transferred performance $P_i$, obtained by fine-tuning $\phi_i$ on the dataset $D$, and evaluating it with a metric, e.g., mean absolute scaled error (MASE), a scale-independent measure (Talagala et al., 2019). A transferability estimation method aims to produce a score $S_i$ for each model $\phi_i$ **without fine-tuning** on dataset $D$. The scores $\{S_i\}_{i=1}^M$ should correlate well with true performance $\{P_i\}_{i=1}^M$, enabling the selection of the best models based on the scores.

As shown in Figure 2, TIMETIC casts transferability estimation as an in-context characteristics-to-performance prediction task. At its core, TIMETIC builds a unified tabular representation that integrates both the data characteristics and the model characteristics. Specifically, time series characterization encodes datasets into a data characteristic table through feature engineering, while model characterization represents TSFMs as a model characteristic table using entropy profiles (detailed in Sections 3.1 and 3.2). Based on these representations, in-context transferability estimation (Section 3.3) proceeds in two stages. In the offline stage, pairs of ground-truth 'characteristics → performance' are collected by fine-tuning to construct an in-context table. In the online stage, this table serves as a context for prompting a tabular foundation model (TabPFN Hollmann et al. (2025) in our case) to learn the mapping between characteristics and performance, allowing accurate estimation of the fine-tuned performance of a target model on a new dataset.

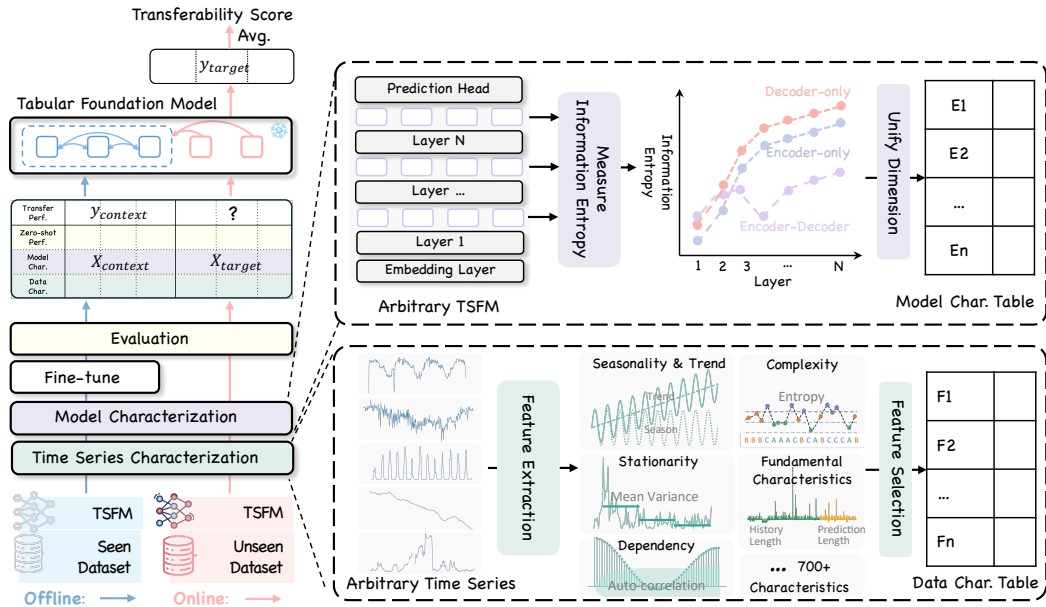

Figure 2: TIMETIC formulates transferability estimation as an in-context characteristics-to-performance prediction task. Dataset characteristics are encoded as a data characteristic table through feature extraction and selection, while models are represented as a model characteristic table using entropy profiles. TIMETIC then operates in two stages: in the offline stage, an in-context table $(X_{context}, y_{context})$ is constructed from characteristic–performance pairs obtained via fine-tuning; in the online stage, this table prompts a tabular foundation model to learn the mapping between characteristics and performance, enabling estimation of a target model's fine-tuned performance $y_{target}$ given a model-data-characteristics table $X_{target}$ in a target dataset. The final transferability score is obtained by averaging the estimated performance across samples.

## 3.1 TIME SERIES CHARACTERIZATION

**Feature extraction** Time series exhibit diverse statistical characteristics that capture their temporal dynamics. For a given dataset $D$, we begin by sampling $n$ time windows $\{\omega_i\}_{i=1}^n$ according to the historical and prediction lengths specified by the forecasting task. For each time window, we extract statistical features as Fulcher (2017); Talagala et al. (2019), using two standard libraries: `tsfresh` (Christ et al., 2018) and `tsfeatures` (Henderson & Fulcher, 2022). The tools can efficiently generate over 700 features that capture diverse properties of time series, including seasonality, stationarity, dependency, complexity, etc. However, these features are highly redundant, which can lead to the curse of dimensionality (Altman & Krzywinski, 2018) and adversely affect characteristic-to-performance regression.

**Feature selection** We perform feature selection guided by the principles of information richness and non-redundancy. To ensure information richness, we select features that minimize the *epistemic uncertainty*, that is, the uncertainty arising from the insufficient observation of the full state of the system. Given some characteristics-performance pairs $\mathcal{T} = (x_i, y_i)_{i>0}$, where $x$ denotes the time series features and $y$ the corresponding transferred model performance, we estimate epistemic uncertainty using TotalVariance ($\mathcal{T}$) as a proxy:

$$\text{TotalVariance}_\phi(\mathcal{T}) = \frac{1}{K} \sum_{k=1}^{K} \text{Var}(y | x \in \mathcal{X}_k) \tag{1}$$

where $\mathcal{X}_1, \ldots, \mathcal{X}_K$ denote the equivalence classes partitioning, i.e., $x, x' \in \mathcal{X}_k$ if and only if $x = x'$ (Akhauri et al., 2025). The variance is then empirically computed over the set of all $y$-values corresponding to inputs within $\mathcal{X}_k$. Intuitively, TotalVariance reflects the distinguishability of features: a smaller value indicates that the feature $x$ provides greater discriminative power to predict $y$. Thus,

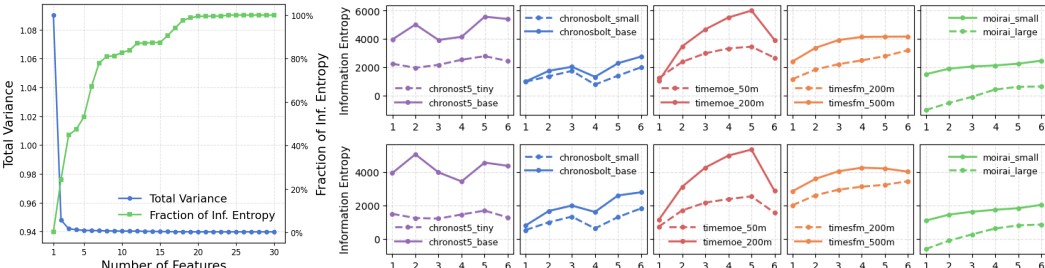

Figure 3: **Left**: TotalVariance significantly declines as the number of features increases, whereas the information content, quantified as the ratio between the joint entropy of a feature subset and that of the full 30-feature set, approaches sufficiency; **Right**: The upper and lower panels show entropy profiles of various TSFMs on the Kdd_cup and Solar datasets. Differences in profile patterns can distinguish model architecture and size: encoder–decoder models (ChronosT5, ChronosBolt) display a two-peak pattern; decoder-only models (TimeMoE, TimesFM) exhibit higher magnitudes than encoder-only models (Moirai); larger hidden dimensionality is associated with higher entropy.

features with lower TotalVariance are more informative for regression. (See Appendix D for a detailed analysis and derivation). In practice, we begin with an empty feature set and iteratively apply a greedy search strategy, adding the feature that minimizes TotalVariance to the set at each step, until the reduction in TotalVariance falls below 0.001. To avoid redundancy, we evaluate the feature set and retain a compact subset that maintains the richness of the information. As shown in Figure 3 (left), the information content of 20 features is comparable to that of the entire 30-feature set. Consequently, we adopt these 20 features with minimal TotalVariance as the final representation for each time series. For a given dataset $D$, this yields a data characteristic table $X_{data} \in \mathbb{R}^{n \times 20}$, where $n$ denotes the number of windows sampled and each row corresponds to the 20-dimensional feature representation of a time window.

## 3.2 MODEL CHARACTERIZATION

Existing approaches to characterize model, such as assigning classification labels (Talagala et al., 2019) or learning model-specific embeddings (Zhang et al., 2023), often struggle to generalize to unseen models, thereby limiting their utility for practical transferability estimation. Inspired by interpretability studies showing that forecast performance correlates with internal representational dimensionality (Kaufman & Azencot, 2024) and that the entropy dynamics across layers reflects key architectural choices (Gabrié et al., 2018; Voita et al., 2019; Ali et al., 2025), we introduce a characterization method based on the trajectory of evolution of the entropy, termed the *entropy profile*. The central premise is that activation functions, operators, parameterization, and hidden dimensions jointly shape value distributions, which in turn determine the magnitude of information entropy. Moreover, entropy can be computed across models without architectural constraints and relies solely on inference statistics, thus entropy profile offers a simple and effective foundation for distinguishing diverse models, without exhaustively accounting for all potential influencing factors.

**Entropy profile** More formally, given a time series represented by $T$ tokens, let $\boldsymbol{t}^i = \{t_1^i, ..., t_T^i\}$ denote the token embeddings after model layer $i$. The entropy profile is defined as follows:

$$\boldsymbol{h} = \bigoplus_{i=1}^{N} \mathcal{H}(\boldsymbol{t}^i),\tag{2}$$

where $N$ is the total number of model layers, $\mathcal{H}$ is the Kozachenko-Leonenko (KL) entropy estimator (Kozachenko, 1987), and $\bigoplus$ denotes concatenation, resulting in $\boldsymbol{h} \in \mathbb{R}^N$. The KL entropy estimator has a critical hyperparameter—the nearest-neighbor count $k$. We adopt a balanced choice of $k = 6$, which mitigates high-variance estimates and reduces instability when computing entropy on high-dimensional feature vectors. For a given dataset $D$ with $n$ sampled time windows $\{\omega_i\}_{i=1}^n$, entropy profiles are computed across windows using at most $10,000$ tokens per layer to compute the information entropy while mitigating computational overhead. To allow comparison between models with different depths, each entropy trajectory is standardized to a fixed length of six—the

minimum depth in our model zoo. Models with more than six layers are compressed by pooling averages in six equal segments, while shallower models are padded by repeating the entropy value of the final layer. Consequently, each model is encoded into a model characteristic table $X_{model} \in \mathbb{R}^{n \times 6}$.

**Entropy profile of TSFMs** Figure 3 (right) presents entropy profiles of TSFMs on the Kdd_cup and Solar datasets. Each model family exhibits a unique profile, with similarities and differences that distinguish models. Within a family, profiles remain consistent across datasets and model sizes, while larger hidden dimensionality is generally associated with higher entropy. Across different families, encoder–decoder architectures (ChronosT5 and ChronosBolt Ansari et al. (2024)) display a distinct entropy drop at the encoder–decoder interface, yielding a two-peak pattern. In contrast, encoder-only models (Moirai Woo et al. (2024)) exhibit lower entropy levels and slower growth across layers compared to decoder-only models (TimeMoE Shi et al. (2025) and TimesFM Das et al. (2023)), a phenomenon attributable to bidirectional attention producing smoother representations.

### 3.3 In-context Transferability Estimation

We reformulate transferability estimation as an in-context characteristics-to-performance prediction task. Specifically, given the observed fine-tuning processes of a TSFM $\phi_i$ on a collection of source datasets $D_{src}$, the goal is to predict the finetuned performance of the model on a downstream target dataset $D_{tgt}$. To this end, TIMETIC performs in-context transferability estimation in two stages: Offline Context Table Construction and Online Target Table Inference, which are detailed as follows:

**Offline context table construction** For each observed finetuning process involving a TSFM $\phi$ and source datasets $D_{src}$, we construct a representation encoding both data and model characteristics, following the procedures described in Section 3.1 and Section 3.2. This yields a data–model characteristic table $X_{context} \in \mathbb{R}^{n \times 26}$, where $n$ denotes the number of time windows sampled from the source datasets, and 26 corresponds to the concatenation of the data characteristics 20 and the characteristics of the model 6. In addition, both the zero-shot and the fine-tuned performance in each time window are appended to the table. The resulting context table is given by $(X_{context}, y_{context}) \in \mathbb{R}^{n \times 28}$, where $y_{context} \in \mathbb{R}^{n \times 1}$ denotes fine-tuned performance. For the cold-start scenario, that is, when no fine-tuned models are available, we can perform fine-tuning on a small number of datasets and encode the results into the context table. This table then serves as a persistent reference to support performance prediction on previously unseen datasets. Importantly, context construction requires only limited offline finetuning on a few datasets, thereby decoupling the one-time finetuning cost from the potentially unbounded number of future target scenarios.

**Online target table inference** Given a target dataset $D_{tgt}$ and a TSFM $\phi_i$ whose transferability is to be estimated, we sample $m$ time windows and construct the target data–model characteristic table $X_{target} \in \mathbb{R}^{m \times 26}$. In the offline stage, a context table $(X_{context}, y_{context})$ is constructed to serve as a structured memory, encoding the mapping between data-model characteristics, zero-shot performance, and fine-tuned performance. By providing both the context table and the target table to a tabular foundation model $\Phi$, predictions of transferred performance on the target dataset can be conditioned on the patterns learned from the context, without requiring gradient updates or retraining. Formally, the estimated transferred performance $y_{target} \in \mathbb{R}^{m \times 1}$ is obtained as

$$y_{target} = \Phi\big(X_{target} \mid (X_{context}, y_{context})\big). \tag{3}$$

The final transferability score $S_i$ of model $\phi_i$ in dataset $D_{tgt}$ is given by the mean of $y_{target}$ in the $m$ sampled time windows.

**Tabular foundation model** In TIMETIC, we employ TabPFN (Hollmann et al., 2025) as the tabular foundation model owing to its strong in-context learning capabilities. TabPFN is a Transformer encoder pre-trained on a large collection of diverse tabular datasets, which enables it to generalize to unseen regression tasks without finetuning. Similar to how large language models leverage in-context examples to perform new tasks, TabPFN can infer task-specific patterns by conditioning on a small number of examples from the target regression problem, and subsequently provide accurate predictions on unseen samples of the same task. This property makes TabPFN particularly well-suited for in-context transferability estimation, as it obviates the need for model retraining and allows flexible organization of context to adapt to diverse transferability estimation scenarios.

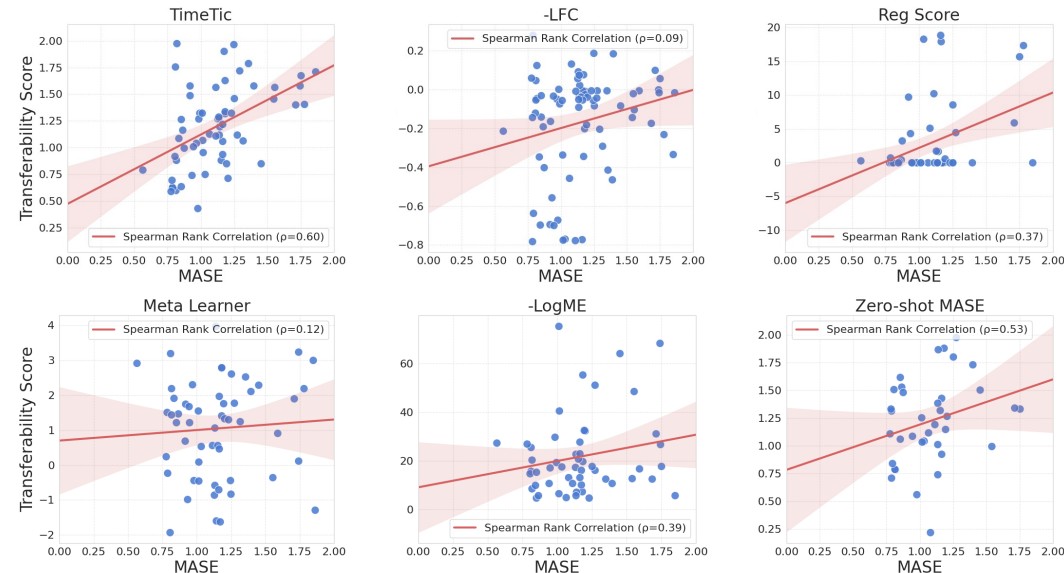

Figure 4: Transferability scores versus actual transferred performance. Each point is a target model's transferability score against its actual transferred performance. More accurate transferability estimation methods show stronger linear and Spearman rank correlations with fine-tuned performance.

## 4 EXPERIMENTS

In Section 4.1, we introduce a benchmark for transferability estimation in TSFMs. Section 4.2 demonstrates the superiority of TIMETIC over existing methods, while Section 4.3 evaluates its generalization in two challenging scenarios: estimating unknown models in seen data, and unknown models on unseen data. Finally, Section 4.4 presents an ablation study on time series characterization, model characterization, and context table size to assess their impact.

### 4.1 TRANSFERABILITY ESTIMATION BENCHMARK

To evaluate transferability estimation methods, we construct a benchmark based on the following five aspects (see Appendix B for details on its construction).

**Target datasets** We use 10 datasets from 4 domains (Nature, Energy, Web and Transport), spanning 5 sampling frequencies (seconds to hours) and 5 key characteristics (trend, seasonality, transition, stationarity and shifting), to ensure the datasets cover diverse temporal patterns.

**Model zoo** 10 models from 5 TSFM families (Chronos, Chronos-Bolt, TimesFM, Moirai, Time-MoE), spanning 10M to 500M parameters, are included to cover various architectures and sizes.

**Ground truth** All TSFMs are fine-tuned on each dataset using unified hyperparameters to establish ground-truth rankings. For each dataset, the last 10% is reserved for testing; the remaining 90% is used for fine-tuning and validation. The rankings are derived through MASE on the test set.

**Transferability estimation baselines** We compare three categories: (i) *Metric-based:* LogME (You et al., 2021), LFC (Tran et al., 2019), and RegScore (Nguyen et al., 2023); (ii) *Meta-learning-based:* a linear meta-estimator adapted from AutoForecast (Abdallah et al., 2022b); (iii) *Zero-shot performance:* using the model's zero-shot performance as the most straightforward proxy.

**Evaluation protocol** Methods are evaluated across short-, medium-, and long-term forecasting tasks under standard and few-shot sampling regimes. The effectiveness is primarily quantified using weighted Kendall's $\tau_w$ between estimated scores $\{S_i\}_{i=1}^M$ and actual finetuned performance $\{P_i\}_{i=1}^M$, with a higher $\tau_w$ indicating more reliable estimate (You et al., 2021; Kazemi et al., 2025).

Table 1: Effectiveness of transferability estimation methods across short-, medium-, and long-horizon forecasting tasks under both standard and few-shot sampling regimes. Reported values are Weighted Kendall's $\tau_w \uparrow$, averaged across 10 datasets.

| Method | Standard | | | Few-shot | | |
|---|---|---|---|---|---|---|
| | short | medium | long | short | medium | long |
| LFC | $-0.114$ | $-0.106$ | $0.101$ | $0.136$ | $0.060$ | $0.102$ |
| LogME | $-0.053$ | $-0.138$ | $-0.138$ | $-0.160$ | $-0.119$ | $-0.176$ |
| RegScore | $-0.272$ | $-0.034$ | $0.018$ | $0.024$ | $0.204$ | $0.187$ |
| Meta learner | $0.053$ | $0.042$ | $-0.089$ | $0.064$ | $0.040$ | $-0.045$ |
| Zero-shot | $0.157$ | $0.329$ | $0.279$ | $0.131$ | $0.262$ | $0.320$ |
| TIMETIC | **0.305** | **0.429** | **0.319** | **0.320** | **0.383** | **0.323** |

## 4.2 PERFORMANCE EVALUATION

**Standard evaluation** We evaluate transferability estimation methods on three forecasting tasks using all time windows from the training set of target datasets. For each model's transferability estimation on a target dataset, TIMETIC leverages a context table that encodes the model's transfer processes on other datasets. As shown in Table 1 (left), TIMETIC consistently outperforms all baselines with higher rank correlations. Although zero-shot performance occasionally aligns with fine-tuned results, it is generally unreliable due to shifts between pretraining and fine-tuning. We also observe that the gap between zero-shot and TIMETIC narrows in long-horizon forecasting, indicating greater challenges in transferability estimation for long-horizon forecasting. Metrics such as RegScore, LogME, and LFC underperform because their assumptions neglect autoregressive error accumulation, while meta-learner–based methods suffer from overfitting and poor generalization. Figure 4 illustrates the transferability scores versus the fine-tuned performance under the medium-horizon task and provides the Spearman rank correlation. Compared to Kendall's $\tau_w$, Spearman's rank correlation emphasizes monotonic consistency; here, TIMETIC achieves the strongest linear alignment with fine-tuned performance and the highest Spearman coefficient of 0.6. In Appendix C, we provide per dataset results and Spearman correlation analyzes.

**Few-shot evaluation.** Few-shot evaluation poses a greater challenge, as only 100 time windows from the training set of the target datasets are used to estimate transferability. With such limited windows, it becomes difficult to fully capture the underlying distribution of a dataset. As shown in Table 1 (right), TIMETIC maintains strong performance with only minor fluctuations in Kendall's $\tau_w$, consistently outperforming all baselines and demonstrating robustness under few-shot settings.

## 4.3 GENERALIZATION EVALUATION

TIMETIC is applicable to a wide range of practical model selection scenarios. It can estimate not only the performance of a known model on a new dataset, but also that of a new model on datasets where other models have already been evaluated. In addition, it can handle the more challenging case of predicting the performance of a new model on entirely unseen datasets.

To simulate these three scenarios, we adopt different constructions of the context table: (i) known target models on unseen datasets - the transfer processes of the target model on the known datasets are encoded in the context table; (ii) unknown target models on seen datasets - other models in the model zoo for the dataset are encoded in the context table; (iii) unknown target models on unseen datasets - other model transfer processes in other datasets are encoded in the context table. As shown in Figure 5, TIMETIC achieves consistently higher rank correlations than relying solely on zero-shot performance in all scenarios. These results highlight the practicality and

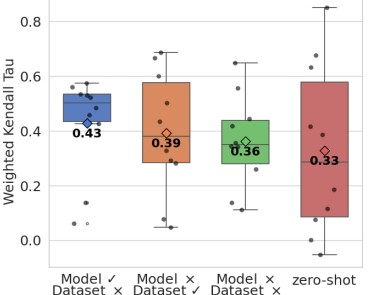

Figure 5: Weighted Kendall's $\tau_w$ of TIMETIC across 10 datasets for different transferability estimation scenarios: (i) known target models on unseen datasets, (ii) unknown target models on seen datasets, and (iii) unknown target models on unseen datasets.

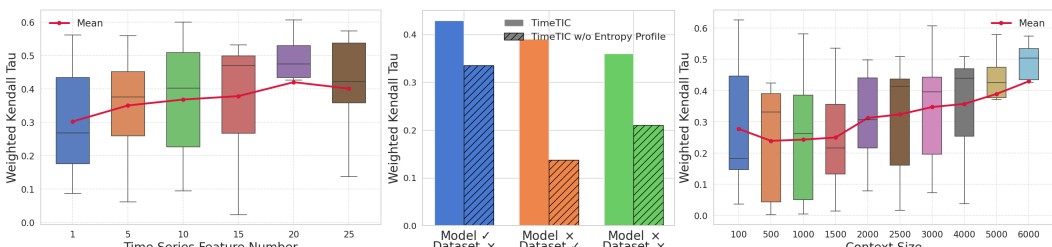

Figure 6: **Left**: Effect of the number of time series features on transferability estimation; **Middle**: Effect of entropy profile on transferability estimation across three scenarios: (i) known target models on unseen datasets, (ii) unknown target models on seen datasets, and (iii) unknown target models on unseen datasets. **Right**: Effect of context table size on transferability estimation.

generalizability of TIMETIC, as it requires only a limited number of observed examples as context to estimate the performance of the unknown model on unseen datasets.

## 4.4 ABLATION STUDY

**Time series feature number** We examine how the number of statistical features impacts TIMETIC. As shown in Figure 6 (left), we incrementally select the first $k$ features that minimize TotalVariance. The results show a consistent improvement as more features are added, since richer representations enhance the discriminative power of the feature space and reduce epistemic uncertainty. However, beyond 20 features, the performance drops slightly, suggesting that additional features introduce redundancy and noise. This observation is consistent with Figure 3 (left), which shows that the information captured by 20 features is nearly equivalent to that of 30 features.

**Model characterization method** Figure 6 (middle) evaluates the contribution of the entropy profile to transferability estimation by comparing TIMETIC with and without it in three scenarios. When the target model is known but the dataset is unseen, the entropy profile yields about 0.1 improvement, indicating that entropy patterns provide useful signals for predicting fine-tuned performance. In more challenging cases, where models are not seen, or both models and datasets are not seen, the entropy profile plays a more critical role, increasing the generalization of TIMETIC by approximately 0.2 and 0.15, respectively. This improvement comes from its ability to capture similarities between models of different architectures or scales, enabling TIMETIC to infer the transferability of unseen models from the transfer processes of known ones.

**Context table size** Another key factor influencing TIMETIC's performance is the size of the context table. Since TIMETIC frames transferability estimation as an in-context characteristic-to-performance prediction task, the size of the context table determines how much prior knowledge can be used for the target prediction. To examine this, we vary the number of time windows most related to the target dataset when constructing the context table and evaluate the impact. As shown in Figure 6 (right), increasing the size of the context from 1,000 to 6,000 substantially improves performance, indicating that richer context information improves TIMETIC. And TIMETIC remains robust even with only 100 time windows. This exhibits TIMETIC's scalability with more known transfer processes and its reliable performance under a limited context.

## 5 CONCLUSION

In this paper, we propose TIMETIC, a novel framework for estimating the transferability of time series foundation models via in-context learning. By encoding model characteristics and data properties into a structured context table, TIMETIC effectively leverages the in-context learning capability of tabular foundation models to provide flexible and accurate performance estimation on unseen datasets. Furthermore, the proposed entropy-profile-based model characterization enhances scalability and generalization, allowing the framework to adapt across diverse transferability estimation scenarios. Comprehensive empirical evaluations demonstrate that TIMETIC consistently surpasses existing methods in model ranking, yielding substantial improvements in correlation with fine-tuned performance. These results establish TIMETIC as a robust and versatile tool for navigating the rapidly expanding landscape of time series foundation models.

**Ethics Statement**

Our research is dedicated exclusively to addressing scientific challenges and does not involve human participants, animals, or materials that pose environmental concerns. We anticipate no ethical risks or conflicts of interest.

**Reproducibility Statement**

We provide the implementation details in Appendices A and B, including method implementations and benchmark construction. The source code and related source of this work are available at `https://anonymous.4open.science/r/ICLR2026-TimeTic-3975` for reproducibility.

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

# TABLE OF CONTENTS

## A  IMPLEMENTATIONS DETAILS

### A.1  FEATURE SELECTION

In this section, we introduce two specific implementations for partitioning equivalence classes in equation 1, along with a greedy search strategy for feature selection, as a supplement to Section 3.1.

**Partitioning of equivalence classes** Given characteristic–performance pairs $\mathcal{T} = (x_i, y_i)_{i>0}$, directly partitioning equivalence classes $\mathcal{X}_k$ based on high-dimensional features $x$ is intractable, as fine-grained clustering becomes unstable in such spaces. To address this, we adopt an approximation procedure combining dimensionality reduction and clustering. Specifically, we first standardize $x$ to zero mean and unit variance, then apply Principal Component Analysis (PCA) and retain the first two components to obtain a reduced feature space. In this space, we cluster the samples into $K$ groups ($K = 100$ in our experiments), with each cluster index serving as a proxy for the equivalence class $\mathcal{X}_k$. Finally, TotalVariance is computed as the average variance across all non-empty clusters, following Equation 1.

**Greedy search strategy** We describe the greedy feature selection algorithm in more detail in Algorithm 1. The algorithm incrementally constructs the feature set by minimizing TotalVariance at each step. This procedure guarantees that each iteration adds the feature that most reduces epistemic uncertainty, until the marginal improvement becomes negligible.

### A.2  TABPFN

In TIMETIC, we adopt TabPFN (Hollmann et al., 2025), a tabular foundation model pretrained on a large collection of regression tasks, as the in-context learner. Both its checkpoint and source code are publicly available. In this section, we provide additional details on TabPFN to help us understand its role within our framework.

**Model architecture** TabPFN treats each cell in a table as a separate position within a sequence. Given a context table and a target table for prediction, all cell values are first normalized using the column-wise mean and standard deviation computed from the context table. These normalized values are then transformed into embeddings through linear projection layers. As illustrated in

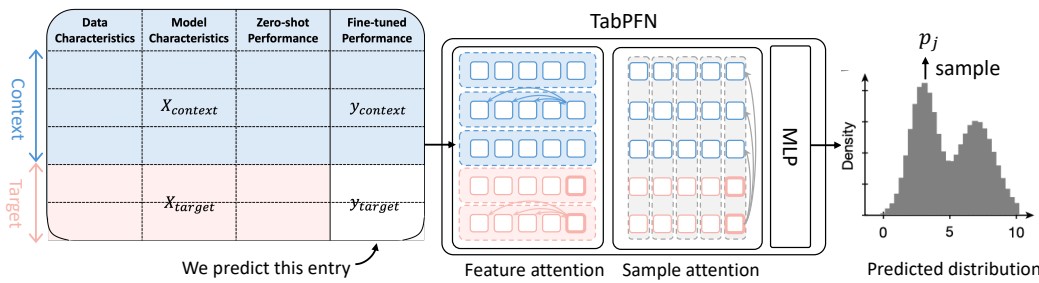

Figure A: The TabPFN-based instance of TIMETIC. We encode observed model behaviors into a context table (shown in blue) and represents new data and models in a target table (shown in red). Then we leverage the in-context learning capabilities of TabPFN to predict the fine-tuned performance on target tasks (denoted as blank cell). TabPFN is an adaptation of the standard Transformer encoder, designed for tabular data using two types of attention mechanisms: one across features and another across samples.

---

**Algorithm 1:** Greedy Feature Selection

---

**Input:** Feature matrix $X \in \mathbb{R}^{n \times d}$, target vector $y \in \mathbb{R}^n$, threshold $\epsilon$
**Output:** Selected feature set $\mathcal{F}_{sel}$

$\mathcal{F}_{sel} \leftarrow \emptyset$;
$\text{TV}_{curr} \leftarrow \inf$;
**repeat**
    best_TV $\leftarrow \text{TV}_{curr}$, $f^* \leftarrow$ None;
    **foreach** $f \notin \mathcal{F}_{sel}$ **do**
        $X_{sel} = \mathcal{F}_{sel}(X) \cup f^*(X)$;
        $\text{TV}_f \leftarrow \text{TotalVariance}(X_{sel}, y)$;
        **if** $TV_f < best\_TV$ **then**
            best_TV $\leftarrow \text{TV}_f$, $f^* \leftarrow f$;

    **if** $f^* \neq$ None **and** $TV_{curr} - best\_TV \geq \epsilon$ **then**
        $\mathcal{F}_{sel} \leftarrow \mathcal{F}_{sel} \cup \{f^*\}$;
        $\text{TV}_{curr} \leftarrow$ best_TV;
**until** *no improvement* $\geq \epsilon$;

---

Figure A, the backbone of TabPFN employs two types of attention mechanisms within each Transformer block: attention across features (columns) and attention across samples (rows), each operating independently along its respective dimension. Finally, TabPFN addresses tabular regression by predicting a probability distribution over possible target values rather than a single point estimate.

**Inference cost** TabPFN is computationally efficient and can be executed on consumer-grade hardware in most scenarios. As reported by Hollmann et al. (2025), for a table with 10,000 rows and 10 columns, TabPFN completes the inference in approximately 0.2 seconds. The computational complexity of the architecture scales quadratically with both the number of samples ($n$) and the number of features ($m$), i.e. $\mathcal{O}(n^2 + m^2)$, while the memory footprint scales linearly with the size of the table, $\mathcal{O}(n + m)$.

## B  BENCHMARK CONSTRUCTION

In this section, we describe the construction of our benchmark, which provides a critical foundation for our experimental analysis. As illustrated in Figure B, the construction pipeline encompasses five key aspects: collection of target datasets and models, unified fine-tuning, selection of baselines, and evaluation protocol. Each of these aspects is elaborated in the following subsections.

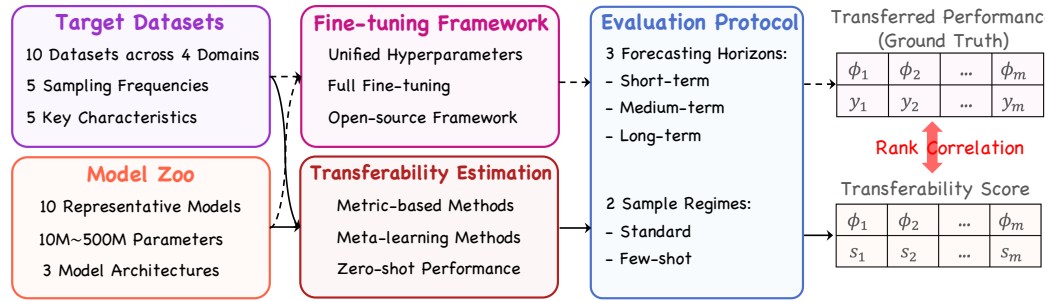

Figure B: Overview of the benchmark construction. To comprehensively evaluate transferability estimation methods for TSFMs, we construct a pipeline (--→) to derive ground-truth transferred performance across 10 datasets, 10 models, and 3 forecasting horizons, under a unified fine-tuning framework. In the evaluation stage (→), we compare TIMETIC against three categories of estimation methods under both standard and few-shot sample regimes, measuring performance by the rank correlation between estimated transferability scores and ground truth.

Table A: Benchmark dataset statistics and forecasting horizons.

| Dataset | Domain | Freq. | #Series | Avg Len | Min Len | Max Len | #Obs | Variates | Short-term | | Med-term | | Long-term | |
|---|---|---|---|---|---|---|---|---|---|---|---|---|---|---|
| | | | | | | | | | Len | Win | Len | Win | Len | Win |
| KDD Cup 2018 (Godahewa et al., 2021) | Nature | H | 270 | 10,898 | 9,504 | 10,920 | 2.94M | 1 | 64 | 20 | 256 | 2 | 512 | 2 |
| Jena Weather (Wu et al., 2021) | Nature | 10T | 1 | 52,704 | 52,704 | 52,704 | 52,704 | 21 | 64 | 20 | 256 | 11 | 512 | 8 |
| ETT2 (Zhou et al., 2020) | Energy | H | 1 | 17,420 | 17,420 | 17,420 | 17,420 | 7 | 64 | 20 | 256 | 4 | 512 | 3 |
| Electricity (Trindade, 2015) | Energy | H | 370 | 35,064 | 35,064 | 35,064 | 12.97M | 1 | 64 | 20 | 256 | 8 | 512 | 5 |
| Solar (Lai et al., 2017) | Energy | H | 137 | 8760 | 8760 | 8760 | 1,200,120 | 1 | 64 | 19 | 256 | 2 | 512 | 8 |
| BizITObs - L2C (Palaskar et al., 2023) | Web/CloudOps | 5T | 1 | 31,968 | 31,968 | 31,968 | 31,968 | 7 | 64 | 20 | 256 | 7 | 512 | 5 |
| Bitbrains - rnd (Shen et al., 2015) | Web/CloudOps | 5T | 500 | 8,640 | 8,640 | 8,640 | 4.32M | 2 | 64 | 18 | 256 | 2 | 512 | 2 |
| BizITObs - App (Palaskar et al., 2023) | Web/CloudOps | 10S | 1 | 8,834 | 8,834 | 8,834 | 8,834 | 2 | 64 | 15 | 256 | 2 | 512 | 1 |
| SZ-Taxi (Wang et al., 2023) | Transport | 15T | 156 | 2,976 | 2,976 | 2,976 | 464,256 | 1 | 64 | 7 | 256 | 1 | 512 | 1 |
| Loop Seattle (Wang et al., 2023) | Transport | 5T | 323 | 105,120 | 105,120 | 105,120 | 33.9M | 1 | 64 | 20 | 256 | 20 | 512 | 15 |

## B.1 TARGET DATASETS

As shown in Table A, our benchmark comprises 10 datasets from four distinct domains, spanning 5 sampling frequencies. These datasets exhibit 5 typical time series characteristics—trend, seasonality, transition, stationarity, and shifting—with example cases illustrated in Figure C. Their diversity simulates real-world TSFM transfer scenarios, providing a solid foundation for evaluating transferability estimation methods.

Following the gift benchmark (Aksu et al., 2024), we define short-, medium- and long-term forecasting tasks to evaluate the transfer performance of TSFM, reflecting the varied forecasting requirements in transfer scenarios. The forecast horizons are set to 64, 256, and 512 time steps, with corresponding context lengths of 256, 1024, and 2048. For each dataset, 90% of the data is used for training and the remaining 10% for testing. During testing, time series are segmented into nonoverlapping windows of length equal to the sum of the context length and forecasting horizon. These settings, along with the number of test windows, are also summarized in Table A.

## B.2 MODEL ZOO

Our benchmark includes a model zoo comprising 10 TSFMs drawn from 5 representative model families, covering a wide spectrum of architectural designs and parameter scales—from 8 million to 500 million parameters. Although all models are based on the Transformer architecture, their performance varies significantly due to differences in encoder-decoder configurations, tokenization schemes, dense versus sparse architectures, and the composition of their pretraining datasets. The

---

[1] https://huggingface.co/collections/Salesforce/moirai-r-models
[2] https://huggingface.co/google/timesfm-1.0-200m
[3] https://huggingface.co/amazon/chronos-t5-tiny
[4] https://huggingface.co/amazon/chronos-bolt-small
[5] https://huggingface.co/Maple728/TimeMoE-50M

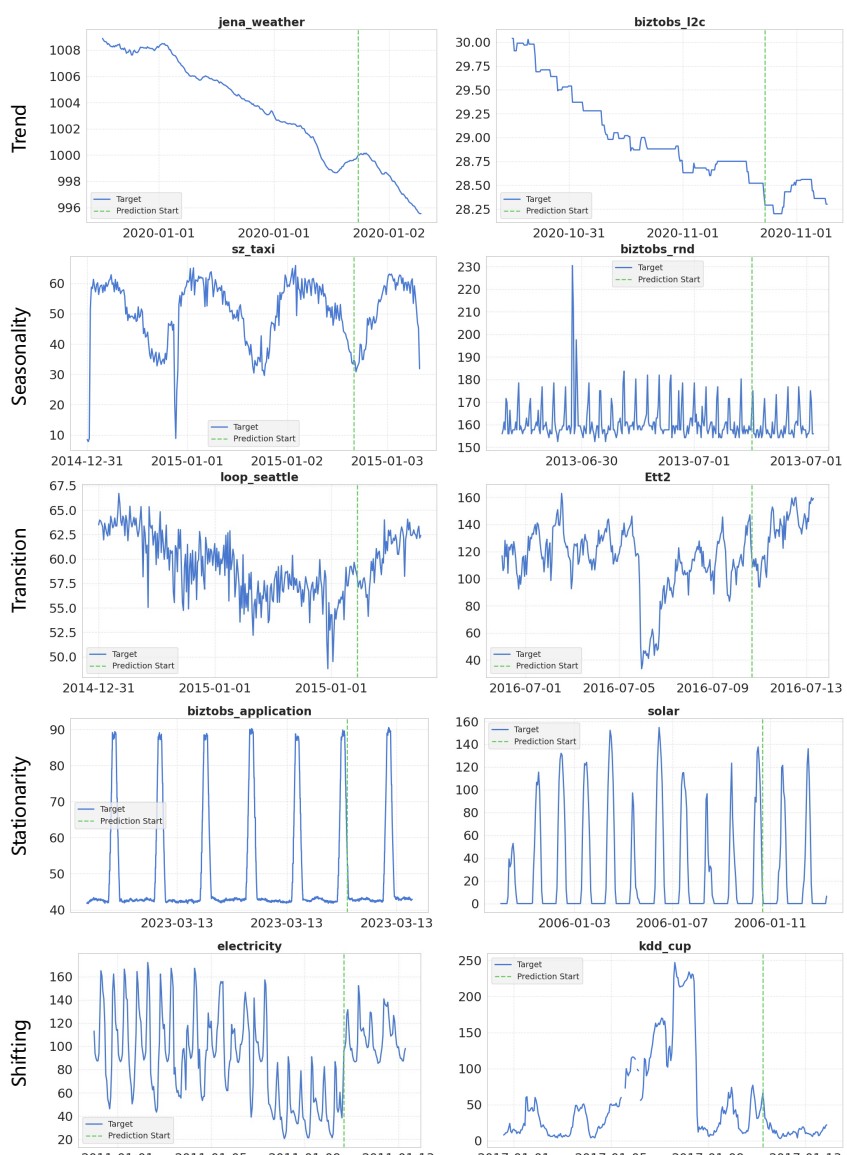

Figure C: 10 datasets illustrating five typical time series characteristics.

Table B: Time series foundation model zoo.

| Model | Architecture | Model Size | | Dataset Size | Input Token | Output Token |
|---|---|---|---|---|---|---|
| Moirai[1] | Encoder-only | 14M | 91M | 231B | Patch | Patch |
| TimesFM[2] | Decoder-only | 200M | 500M | 100B | Patch | Patch |
| Chronos[3] | Enc-Dec | 8M | 20M | 84B | Point | Point |
| Chronos-bolt[4] | Enc-Dec | 48M | 205M | 84B | Patch | Patch |
| Time-MoE[5] | Decoder-only | 50M | 200M | 309B | Point | Patch |

characteristics of the models in our zoo are summarized in Table B, and a brief introduction to each model family is provided below:

**Moirai** (Woo et al., 2024) is an encoder-only Transformer that uses adaptive patch tokenization to accommodate time series with varying frequencies, along with a flexible attention mechanism to support multivariate inputs. It also features a patch-wise parameterized prediction head for distribu-

Table C: Ground truth finetuned performance of various time series foundation models in short-term, medium-term, and long-term forecasting tasks.

| Dataset | Chronos-tiny | Chronos-base | Chronos-bolt-base | Chronos-bolt-small | Moirai-large | Moirai-small | Time-MoE-50M | Time-MoE-200M | TimesFM-200M | TimesFM-500M |
|---|---|---|---|---|---|---|---|---|---|---|
| **Short-term forecasting tasks** | | | | | | | | | | |
| kdd_cup_2018_with_missing:H | 1.085 | 0.997 | 0.763 | 0.850 | 1.025 | 1.004 | 0.916 | 0.993 | 1.092 | 0.972 |
| jena_weather:10T | 2.314 | 1.752 | 1.557 | 1.515 | 1.905 | 1.523 | 1.422 | 1.285 | 1.229 | 1.158 |
| ett2:H | 1.161 | 1.194 | 1.036 | 1.007 | 1.064 | 1.090 | 1.085 | 1.093 | 1.134 | 1.084 |
| electricity:H | 1.135 | 0.945 | 0.941 | 0.872 | 1.062 | 1.203 | 0.938 | 0.947 | 1.355 | 1.206 |
| solar:H | 1.412 | 1.322 | 1.367 | 1.456 | 1.447 | 1.493 | 1.332 | 1.419 | 2.359 | 1.618 |
| bizitobs_l2c:5T | 0.571 | 0.618 | 0.568 | 0.564 | 0.613 | 0.568 | 0.598 | 0.611 | 0.621 | 0.564 |
| bitbrains_rnd:5T | 2.276 | 2.155 | 2.053 | 2.163 | 2.392 | 2.884 | 2.059 | 2.008 | 2.635 | 2.473 |
| bizitobs_application | 2.847 | 3.176 | 2.842 | 2.805 | 2.734 | 3.246 | 2.719 | 2.681 | 2.917 | 3.164 |
| SZ_TAXI:15T | 0.884 | 0.877 | 0.828 | 0.819 | 0.808 | 0.843 | 0.807 | 0.813 | 0.812 | 0.818 |
| LOOP_SEATTLE:5T | 0.733 | 0.711 | 0.689 | 0.650 | 0.672 | 0.660 | 0.626 | 0.621 | 0.873 | 0.876 |
| **Medium-term forecasting tasks** | | | | | | | | | | |
| kdd_cup_2018_with_missing:H | 1.483 | 1.240 | 0.706 | 0.812 | 1.158 | 1.103 | 1.164 | 1.145 | 1.099 | 1.034 |
| jena_weather:10T | 1.610 | 1.258 | 0.977 | 0.944 | 1.208 | 0.998 | 1.180 | 1.123 | 1.123 | 0.831 |
| ett2:H | 1.474 | 1.219 | 1.021 | 1.046 | 1.043 | 1.059 | 1.186 | 1.096 | 1.169 | 1.164 |
| electricity:H | 1.303 | 1.130 | 1.040 | 1.020 | 1.096 | 1.222 | 1.297 | 1.262 | 1.364 | 1.285 |
| solar:H | 1.270 | 0.968 | 1.153 | 1.262 | 1.169 | 1.118 | 0.767 | 0.871 | 1.694 | 1.227 |
| bizitobs_l2c:5T | 1.147 | 1.125 | 1.222 | 1.128 | 0.991 | 1.003 | 1.688 | 1.661 | 1.331 | 1.184 |
| bitbrains_rnd:5T | 1.895 | 1.742 | 1.419 | 1.702 | 2.076 | 2.818 | 3.501 | 2.743 | 2.306 | 2.107 |
| bizitobs_application | 12.494 | 9.412 | 1.765 | 1.867 | 2.314 | 8.932 | 2.750 | 2.046 | 6.429 | 7.151 |
| SZ_TAXI:15T | 0.901 | 0.914 | 0.816 | 0.804 | 0.797 | 0.817 | 0.827 | 0.821 | 0.843 | 0.815 |
| LOOP_SEATTLE:5T | 1.152 | 0.857 | 0.890 | 0.850 | 0.798 | 0.753 | 1.175 | 0.970 | 0.928 | 0.973 |
| **Long-term forecasting tasks** | | | | | | | | | | |
| kdd_cup_2018_with_missing:H | 1.778 | 1.291 | 0.850 | 0.942 | 1.193 | 1.137 | 1.395 | 1.249 | 1.229 | 1.134 |
| jena_weather:10T | 2.172 | 1.387 | 1.202 | 1.152 | 1.451 | 1.163 | 1.749 | 1.710 | 1.271 | 1.080 |
| ett2:H | 2.145 | 2.043 | 1.010 | 1.181 | 1.112 | 1.171 | 2.436 | 2.062 | 1.179 | 1.171 |
| electricity:H | 1.552 | 1.314 | 1.132 | 1.132 | 1.272 | 1.347 | 3.696 | 3.162 | 1.683 | 1.540 |
| solar:H | 1.355 | 0.916 | 1.031 | 1.161 | 1.015 | 1.109 | 0.843 | 0.946 | 1.848 | 1.182 |
| bizitobs_l2c:5T | 1.140 | 1.127 | 0.783 | 0.804 | 0.562 | 0.966 | 0.992 | 0.982 | 1.246 | 1.138 |
| bitbrains_rnd:5T | 1.861 | 1.545 | 1.161 | 1.181 | 1.740 | 2.181 | 1.590 | 1.742 | 2.104 | 1.836 |
| bizitobs_application | 9.969 | 9.745 | 2.274 | 2.712 | 3.680 | 9.136 | 5.120 | 3.313 | 8.389 | 9.672 |
| SZ_TAXI:15T | 0.874 | 0.932 | 0.810 | 0.816 | 0.776 | 0.787 | 0.817 | 0.809 | 0.834 | 0.792 |
| LOOP_SEATTLE:5T | 1.251 | 0.919 | 0.864 | 0.851 | 0.977 | 0.785 | 1.065 | 1.012 | 0.974 | 0.895 |

tional forecasting. In our experiments, we include Moirai-small (14M) and Moirai-base (91M) as candidate models.

**TimesFM** (Das et al., 2023) is a decoder-only Transformer tailored for time series forecasting. It extends the standard decoder-only architecture by adopting patch-based tokenization and detokenization strategies, allowing it to effectively handle time series inputs and generate forecasts. We include TimesFM-200M and TimesFM-500M in our candidate models.

**Chronos** (Ansari et al., 2024) is an LLM-based TSFM that repurposes the T5 encoder-decoder architecture for time series forecasting. Instead of using T5's original text-based tokenizer, Chronos applies value quantization and dequantization to convert the regression task into a classification problem. It is pretrained on a large-scale time series corpus comprising 84 billion time points. Chronos-tiny (8M) and Chronos-min (20M) are included in our candidate models.

**Chronos-bolt** (Ansari et al., 2024) also builds on the T5 architecture but introduces significant differences in tokenization and prediction strategies. It employs patch-based tokenization and replaces autoregressive decoding with single-pass inference, predicting a fixed-length patch in each pass. For longer forecasting horizons, it iteratively encodes the historical context and predicts a future patch. We include Chronos-bolt-small (48M) and Chronos-bolt-base (205M) in our model zoo.

**Time-MoE** (Shi et al., 2025) is a sparse decoder-only Transformer incorporating a mixture-of-experts (MoE) architecture to enable scalable time series forecasting. By leveraging sparse routing instead of a fully dense structure, Time-MoE scales effectively with minimal computational overhead. It also uses point-wise embeddings and multi-scale patch-based predictions. We select Time-MoE with two different sizes (50M and 200M) for inclusion in our model zoo.

### B.3   GROUND TRUTH

To evaluate transferability estimation approaches, we fine-tune all models to obtain their actual fine-tuned performance and ranking. A unified fine-tuning strategy is applied across all models to eliminate variability introduced by the fine-tuning process itself, ensuring a fair comparison of their transferability.

We choose to fine-tune all parameters of each model, which is a simple but general approach. Each model is fine-tuned for 1 epoch using a batch size of 32 and a maximum sequence length of 2560. Optimization is performed with the AdamW optimizer and a constant learning rate of 1e-5. The

final checkpoint after 1 epoch is reserved for final evaluation on the test set to determine the actual fine-tuned performance. All fine-tuning experiments are conducted on a single H100 GPU. The actual fine-tuned results under the three forecasting tasks are reported in Table C.

## B.4 Baselines

**LFC** (Tran et al., 2019) adopts a linearized framework to approximate fine-tuning and measures the Label-Feature Correlation to estimate transferability. We compute the mean LFC across all token embeddings produced by the model backbone within the forecasting horizon, and use it as the transferability score for each sample.

**LogME** (You et al., 2021) models transferability through estimating the maximum value of the target label evidence given the target features extracted from the pre-trained model. We also compute the mean LogME across all token embeddings produced by the model backbone within the forecasting horizon, and use this as the transferability score for a given sample.

**RegScore** (Nguyen et al., 2023) assesses transferability by measuring the error of a linear regression model trained to predict labels from features. We compute the RegScore between all token embeddings produced by the model backbone within the forecasting horizon and their corresponding labels, and use the mean value as the transferability score for each sample.

**Meta-learner**. The general meta-learner in AutoForecast Abdallah et al. (2022a) is a linear model designed to project dataset meta-features to model performance. In our experiments, we adapt this meta-learner to predict fine-tuned performance based on data characteristics, model entropy profile, and zero-shot performance. The training data is identical to the corpus collected for TIMETIC.

**Zero-shot** performance is the simplest proxy for estimating TSFM's transferability. We use the MASE to measure the zero-shot performance on a sample and use it as the transferability score.

## B.5 Evaluation Metrics

**Weighted Kendall's tau** ($\tau_w$) is a statistic that measures the ordinal association between two ranked lists while assigning different importance to item pairs. It is defined as:

$$\tau_w = 1 - \frac{2 \sum_{(i,j):i<j} w_{ij} \cdot \mathbb{I}\big[(x_i - x_j)(y_i - y_j) < 0\big]}{\sum_{(i,j):i<j} w_{ij}}$$

where $w_{ij}$ is a nonnegative weight assigned to the pair $(i, j)$, and $\mathbb{I}[\cdot]$ is the indicator function that equals 1 if the pair is discordant and 0 otherwise. By weighting different item pairs, $\tau_w$ allows emphasizing errors at the top of the ranking or other positions of interest. The value of $\tau_w$ ranges from $-1$ (inverse ranking) to 1 (perfect agreement), with 0 indicating no ordinal correlation. Compared with the standard Kendall's tau, the weighted version provides greater flexibility in applications where certain ranking positions are more critical than others.

**Spearman's rank correlation** ($\rho$) is a nonparametric statistic that measures the monotonic association between two ranked lists. It is defined as:

$$\rho = 1 - \frac{6 \sum_{i=1}^{n} d_i^2}{n(n^2 - 1)}$$

where $d_i$ is the difference between the ranks of the $i$-th item in the two lists, and $n$ is the total number of items being ranked. The value of $\rho$ ranges from $-1$ (perfect inverse monotonic relationship) to 1 (perfect monotonic agreement), with 0 indicating no monotonic correlation. Compared with Kendall's tau, Spearman's $\rho$ is based on rank differences rather than concordant and discordant pairs, making it computationally simpler for large $n$.

**Mean Absolute Scaled Error** (MASE) evaluates forecast accuracy by comparing it to a naive baseline. It is defined as:

$$\text{MASE} = \frac{\frac{1}{T}\sum_{t=1}^{T}|y_t - \hat{y}_t|}{\frac{1}{T-m}\sum_{t=m+1}^{T}|y_t - y_{t-m}|}$$

where $y_t$ is the true value, $\hat{y}_t$ is the predicted value, $T$ is the length of the forecast period, and $m$ is the seasonality of the series (with $m = 1$ for non-seasonal data). The denominator represents the in-sample mean absolute error of a naive forecasting method (e.g., seasonal naive). MASE is scale-free and interpretable: a value less than 1 indicates the model outperforms the naive baseline.

## C  ADDITIONAL EXPERIMENTAL RESULTS

### C.1  PERFORMANCE EVALUATION USING WEIGHTED KENDALL TAU

Tables D and E report the performance of transferability estimation methods in short-, medium-, and long-term forecasting tasks in the standard and few-shot regimes. TIMETIC achieves the highest correlations on most datasets, consistently outperforming all baselines. We also observed fluctuations in transferability estimation performance across different forecast horizons within the same data set, suggesting that the forecast horizon is an important factor influencing TSFM performance and ranking. Moreover, dataset characteristics introduce varying challenges: for example, TIMETIC performs poorly on the sz_taxi dataset but consistently achieves strong results on the bitbrains_rnd dataset.

### C.2  PERFORMANCE EVALUATION USING SPEARMAN CORRELATION

Tables F and G report the Spearman rank correlations of transferability estimation methods across short-, medium-, and long-term forecasting tasks under both standard and few-shot regimes. Unlike weighted Kendall's $\tau_w$, which emphasizes pairwise concordance with importance weights, Spearman correlation evaluates the global monotonic relationship between two rankings, making it more sensitive to overall rank consistency. From the results, we observe that zero-shot performance provides a relatively strong baseline with higher correlation than other metrics. By incorporating richer time series features and model characterization, TIMETIC achieves about a 30% improvement over zero-shot performance on average.

### C.3  DISCUSSIONS ON ENTROPY PROFILE

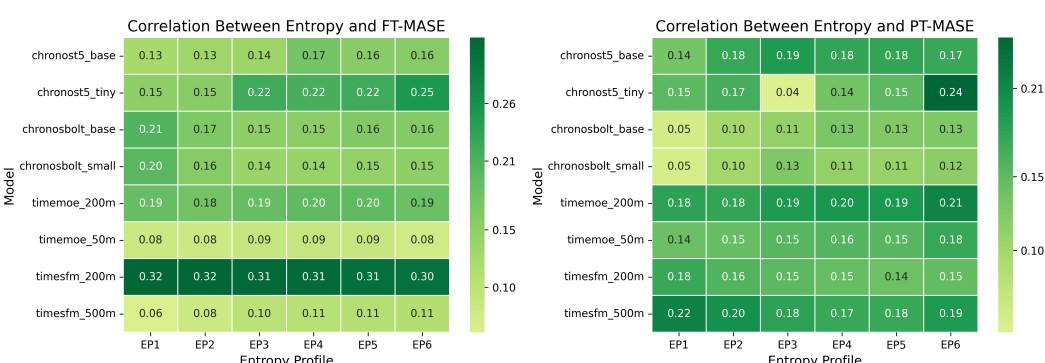

Figure D: Pearson correlation between each entropy-profile dimension and both finetuned performance (**left**) and zero-shot performance (**right**). Overall, the entropy profile shows a positive correlation with both finetuned and zero-shot performance. Zero-shot performance tends to correlate more strongly with the cross-entropy of deeper-layer features, whereas this layer-dependent pattern is less evident for finetuned performance.

**Correlation between entropy profile and performance.** Figure D shows the Pearson correlation between each entropy-profile dimension and both finetuned and zero-shot performance. The

Table D: Performance comparison of transferability estimation methods for short-term, medium-term, and long-term forecasting under standard evaluation.

| Method | Downstream Target Datasets | | | | | | | | | | Mean |
|---|---|---|---|---|---|---|---|---|---|---|---|
| | kdd_cup | bizitobs_l2c | electricity | solar | sz_taxi | jena_weather | ett2 | bitbrains_rnd | bizitobs_app | loop_seattle | |
| **Short-term forecasting** | | | | | | | | | | | |
| LFC | 0.036 | -0.038 | -0.437 | 0.618 | -0.448 | -0.605 | -0.471 | -0.441 | 0.007 | 0.638 | -0.114 |
| LogME | 0.432 | -0.245 | 0.040 | 0.519 | -0.556 | -0.093 | -0.528 | 0.016 | 0.162 | -0.272 | -0.053 |
| RegScore | -0.354 | -0.178 | -0.301 | -0.677 | 0.069 | -0.274 | -0.510 | 0.041 | -0.294 | -0.246 | -0.272 |
| Meta learner | -0.281 | 0.339 | 0.221 | 0.266 | 0.304 | -0.120 | 0.260 | -0.473 | -0.149 | 0.159 | 0.053 |
| Zero-shot | 0.406 | -0.044 | -0.253 | 0.444 | 0.038 | 0.411 | -0.157 | 0.144 | 0.110 | 0.471 | 0.157 |
| TIMETIC | 0.463 | 0.320 | 0.218 | 0.159 | 0.152 | 0.372 | 0.190 | 0.606 | 0.112 | 0.456 | 0.305 |
| **Medium-term forecasting** | | | | | | | | | | | |
| LFC | -0.130 | 0.016 | -0.301 | 0.510 | -0.289 | -0.435 | -0.394 | -0.296 | 0.016 | 0.402 | -0.106 |
| LogME | -0.205 | 0.411 | 0.119 | -0.147 | -0.328 | -0.169 | -0.631 | -0.474 | 0.411 | 0.001 | -0.138 |
| RegScore | 0.200 | -0.131 | -0.296 | -0.226 | 0.491 | 0.135 | 0.317 | -0.105 | 0.274 | -0.320 | 0.034 |
| Meta learner | 0.680 | -0.320 | -0.015 | 0.105 | -0.436 | 0.205 | 0.260 | 0.266 | -0.504 | 0.177 | 0.042 |
| Zero-shot | 0.386 | -0.053 | 0.075 | 0.187 | 0.632 | 0.850 | 0.678 | 0.002 | 0.417 | 0.115 | 0.329 |
| TIMETIC | 0.137 | 0.522 | 0.426 | 0.574 | 0.061 | 0.561 | 0.536 | 0.530 | 0.485 | 0.459 | 0.429 |
| **Long-term forecasting** | | | | | | | | | | | |
| LFC | -0.079 | 0.385 | 0.005 | 0.597 | -0.499 | -0.317 | 0.234 | -0.102 | 0.330 | 0.451 | 0.101 |
| LogME | -0.283 | -0.511 | -0.052 | -0.256 | -0.411 | 0.354 | -0.254 | -0.321 | 0.346 | 0.013 | -0.138 |
| RegScore | 0.307 | -0.146 | -0.606 | -0.340 | 0.717 | 0.264 | 0.334 | -0.295 | 0.241 | -0.300 | 0.018 |
| Meta learner | 0.411 | -0.119 | 0.105 | -0.221 | -0.437 | -0.467 | 0.008 | 0.105 | 0.084 | -0.361 | -0.089 |
| Zero-shot | 0.393 | 0.518 | -0.079 | 0.099 | 0.489 | 0.346 | 0.251 | -0.013 | 0.547 | 0.242 | 0.279 |
| TIMETIC | 0.215 | 0.632 | 0.197 | 0.334 | 0.052 | 0.037 | 0.327 | 0.632 | 0.445 | 0.038 | 0.319 |

Table E: Performance comparison of transferability estimation methods for short-term, medium-term, and long-term forecasting under few-shot evaluation.

| Method | Downstream Target Datasets | | | | | | | | | | Mean |
|---|---|---|---|---|---|---|---|---|---|---|---|
| | kdd_cup | bizitobs_l2c | electricity | solar | sz_taxi | jena_weather | ett2 | bitbrains_rnd | bizitobs_app | loop_seattle | |
| **Short-term forecasting (few-shot)** | | | | | | | | | | | |
| LFC | 0.316 | 0.266 | 0.282 | 0.628 | -0.182 | -0.631 | 0.187 | -0.180 | 0.124 | 0.551 | 0.136 |
| LogME | 0.080 | 0.114 | -0.033 | 0.257 | -0.559 | 0.067 | -0.626 | -0.199 | -0.432 | -0.268 | -0.160 |
| RegScore | -0.215 | -0.254 | 0.001 | -0.166 | 0.175 | 0.245 | 0.357 | 0.383 | -0.104 | -0.179 | 0.024 |
| Meta learner | -0.366 | 0.277 | 0.221 | 0.266 | 0.263 | -0.120 | 0.374 | -0.367 | -0.184 | 0.272 | 0.064 |
| Zero-shot | 0.019 | -0.144 | -0.078 | 0.445 | 0.145 | 0.350 | 0.157 | -0.051 | 0.119 | 0.346 | 0.131 |
| TIMETIC | 0.538 | 0.286 | 0.285 | 0.316 | 0.134 | 0.293 | 0.107 | 0.451 | 0.442 | 0.241 | 0.320 |
| **Medium-term forecasting (few-shot)** | | | | | | | | | | | |
| LFC | 0.110 | 0.140 | -0.186 | 0.686 | -0.133 | -0.288 | -0.044 | -0.039 | 0.016 | 0.339 | 0.060 |
| LogME | -0.143 | 0.487 | -0.255 | -0.256 | -0.328 | -0.198 | -0.605 | -0.314 | 0.411 | 0.007 | -0.119 |
| RegScore | 0.530 | 0.436 | -0.132 | -0.277 | 0.481 | 0.326 | 0.505 | 0.095 | 0.274 | -0.203 | 0.204 |
| Meta learner | 0.680 | -0.184 | -0.015 | 0.105 | -0.436 | 0.455 | 0.207 | -0.081 | -0.505 | 0.177 | 0.040 |
| Zero-shot | 0.508 | 0.067 | 0.075 | 0.186 | 0.405 | 0.781 | -0.081 | 0.047 | 0.417 | 0.213 | 0.262 |
| TIMETIC | 0.137 | 0.451 | 0.338 | 0.593 | 0.061 | 0.527 | 0.436 | 0.340 | 0.485 | 0.459 | 0.383 |
| **Long-term forecasting (few-shot)** | | | | | | | | | | | |
| LFC | 0.052 | 0.324 | 0.310 | 0.594 | -0.528 | -0.280 | 0.265 | -0.433 | 0.330 | 0.384 | 0.102 |
| LogME | -0.283 | -0.415 | 0.019 | -0.374 | -0.411 | 0.015 | -0.144 | -0.429 | 0.346 | -0.083 | -0.176 |
| RegScore | 0.361 | -0.174 | -0.546 | -0.453 | 0.612 | 0.175 | 0.442 | -0.518 | 0.241 | 0.046 | 0.019 |
| Meta learner | 0.411 | -0.119 | 0.105 | -0.221 | -0.437 | 0.095 | -0.040 | 0.089 | 0.084 | -0.414 | -0.045 |
| Zero-shot | 0.425 | 0.536 | 0.001 | 0.152 | 0.508 | 0.408 | 0.376 | 0.071 | 0.547 | 0.173 | 0.320 |
| TIMETIC | 0.305 | 0.672 | 0.197 | 0.334 | 0.088 | 0.226 | 0.469 | 0.458 | 0.445 | 0.046 | 0.323 |

Table F: Spearman ranking correlation of transferability estimation methods for short-term, medium-term, and long-term forecasting under standard evaluation.

| Method | Downstream Target Datasets | | | | | | | | | | Mean |
|---|---|---|---|---|---|---|---|---|---|---|---|
| | kdd_cup | bizitobs_l2c | electricity | solar | sz_taxi | jena_weather | ett2 | bitbrains_rnd | bizitobs_app | loop_seattle | |
| **Short-term forecasting** | | | | | | | | | | | |
| LFC | 0.261 | -0.079 | -0.539 | 0.467 | -0.624 | -0.576 | -0.624 | -0.479 | 0.152 | 0.612 | -0.143 |
| LogME | 0.358 | -0.042 | 0.152 | 0.770 | -0.673 | -0.273 | -0.612 | 0.394 | 0.115 | -0.527 | -0.034 |
| RegScore | -0.491 | 0.055 | -0.479 | -0.745 | 0.285 | -0.018 | -0.648 | 0.006 | -0.236 | -0.345 | -0.262 |
| Meta learner | -0.273 | 0.624 | 0.309 | 0.079 | 0.273 | -0.188 | 0.358 | -0.685 | -0.139 | -0.164 | 0.019 |
| Zero-shot | 0.588 | -0.067 | -0.212 | 0.564 | -0.006 | 0.527 | 0.200 | 0.188 | 0.139 | 0.648 | 0.257 |
| TIMETIC | 0.661 | 0.291 | 0.394 | 0.067 | 0.176 | 0.261 | 0.103 | 0.503 | 0.345 | 0.733 | 0.353 |
| **Medium-term forecasting** | | | | | | | | | | | |
| LFC | -0.212 | 0.103 | -0.333 | 0.442 | -0.224 | -0.479 | -0.503 | -0.261 | 0.273 | 0.515 | -0.068 |
| LogME | -0.358 | 0.115 | 0.018 | -0.127 | -0.394 | -0.297 | -0.794 | -0.491 | 0.236 | 0.236 | -0.185 |
| RegScore | 0.188 | -0.224 | -0.419 | -0.176 | 0.261 | 0.236 | 0.164 | 0.006 | 0.370 | -0.285 | 0.012 |
| Meta learner | 0.624 | -0.164 | 0.030 | -0.176 | -0.467 | 0.176 | 0.382 | 0.345 | -0.721 | -0.091 | -0.006 |
| Zero-shot | 0.648 | 0.078 | 0.212 | 0.030 | 0.794 | 0.903 | 0.697 | 0.358 | 0.515 | 0.430 | 0.467 |
| TIMETIC | 0.521 | 0.697 | 0.682 | 0.539 | 0.394 | 0.582 | 0.733 | 0.555 | 0.697 | 0.600 | 0.600 |
| **Long-term forecasting** | | | | | | | | | | | |
| LFC | -0.103 | 0.685 | -0.030 | 0.394 | -0.467 | -0.297 | 0.042 | -0.273 | 0.697 | 0.358 | 0.101 |
| LogME | -0.345 | -0.733 | -0.067 | -0.224 | -0.370 | 0.491 | -0.358 | -0.333 | 0.176 | -0.018 | -0.178 |
| RegScore | 0.236 | -0.236 | -0.657 | -0.333 | 0.612 | 0.273 | 0.612 | -0.273 | 0.176 | -0.455 | -0.004 |
| Meta learner | 0.564 | -0.152 | 0.345 | -0.261 | -0.358 | -0.382 | 0.321 | 0.006 | -0.139 | -0.552 | -0.061 |
| Zero-shot | 0.684 | 0.455 | 0.176 | -0.055 | 0.539 | 0.552 | 0.527 | 0.309 | 0.321 | 0.297 | 0.381 |
| TIMETIC | 0.527 | 0.830 | 0.552 | 0.078 | 0.285 | 0.079 | 0.539 | 0.673 | 0.539 | 0.079 | 0.418 |

Table G: Spearman ranking correlation of transferability estimation methods for short-term, medium-term, and long-term forecasting under few-shot evaluation.

| Method | Downstream Target Datasets | | | | | | | | | | Mean |
|---|---|---|---|---|---|---|---|---|---|---|---|
| | kdd_cup | bizitobs_l2c | electricity | solar | sz_taxi | jena_weather | ett2 | bitbrains_rnd | bizitobs_app | loop_seattle | |
| **Short-term forecasting (few-shot)** | | | | | | | | | | | |
| LFC | 0.467 | 0.382 | 0.333 | 0.479 | -0.176 | -0.770 | 0.236 | -0.164 | -0.224 | 0.539 | 0.110 |
| LogME | -0.103 | 0.333 | -0.261 | 0.430 | -0.624 | -0.115 | -0.721 | -0.321 | -0.600 | -0.588 | -0.257 |
| RegScore | -0.006 | -0.115 | 0.103 | -0.236 | 0.091 | 0.394 | 0.297 | 0.543 | -0.079 | -0.309 | 0.068 |
| Meta learner | -0.321 | 0.515 | 0.309 | 0.079 | 0.236 | -0.188 | 0.394 | -0.636 | -0.164 | -0.018 | 0.021 |
| Zero-shot | 0.248 | -0.139 | 0.103 | 0.612 | 0.188 | 0.430 | 0.297 | -0.042 | 0.297 | 0.370 | 0.236 |
| TIMETIC | 0.576 | 0.394 | 0.394 | 0.479 | 0.152 | 0.370 | 0.139 | 0.648 | 0.552 | 0.291 | 0.399 |
| **Medium-term forecasting (few-shot)** | | | | | | | | | | | |
| LFC | 0.224 | 0.418 | -0.212 | 0.539 | -0.103 | -0.418 | -0.321 | -0.055 | 0.273 | 0.479 | 0.082 |
| LogME | -0.248 | 0.515 | -0.333 | -0.297 | -0.394 | -0.321 | -0.758 | -0.261 | 0.236 | 0.212 | -0.165 |
| RegScore | 0.600 | 0.612 | 0.025 | -0.115 | 0.273 | 0.321 | 0.442 | 0.030 | 0.370 | -0.139 | 0.242 |
| Meta learner | 0.624 | 0.006 | 0.030 | -0.176 | -0.467 | 0.285 | 0.236 | 0.018 | -0.721 | -0.091 | -0.025 |
| Zero-shot | 0.660 | 0.042 | 0.212 | 0.030 | 0.648 | 0.855 | 0.006 | 0.248 | 0.515 | 0.576 | 0.379 |
| TIMETIC | 0.321 | 0.624 | 0.285 | 0.588 | 0.394 | 0.345 | 0.515 | 0.321 | 0.697 | 0.600 | 0.469 |
| **Long-term forecasting (few-shot)** | | | | | | | | | | | |
| LFC | -0.042 | 0.661 | 0.127 | 0.382 | -0.394 | -0.224 | 0.091 | -0.588 | 0.697 | 0.345 | 0.105 |
| LogME | -0.345 | -0.636 | -0.079 | -0.394 | -0.370 | 0.127 | -0.273 | -0.539 | 0.176 | -0.091 | -0.242 |
| RegScore | 0.552 | -0.188 | -0.644 | -0.467 | 0.430 | 0.370 | 0.685 | -0.673 | 0.176 | 0.200 | 0.044 |
| Meta learner | 0.564 | -0.152 | 0.345 | -0.261 | -0.358 | 0.067 | 0.224 | 0.006 | -0.139 | -0.588 | -0.029 |
| Zero-shot | 0.697 | 0.455 | 0.273 | 0.006 | 0.564 | 0.685 | 0.648 | 0.248 | 0.321 | 0.236 | 0.413 |
| TIMETIC | 0.539 | 0.842 | 0.552 | 0.079 | 0.273 | 0.188 | 0.576 | 0.539 | 0.539 | 0.418 | 0.451 |

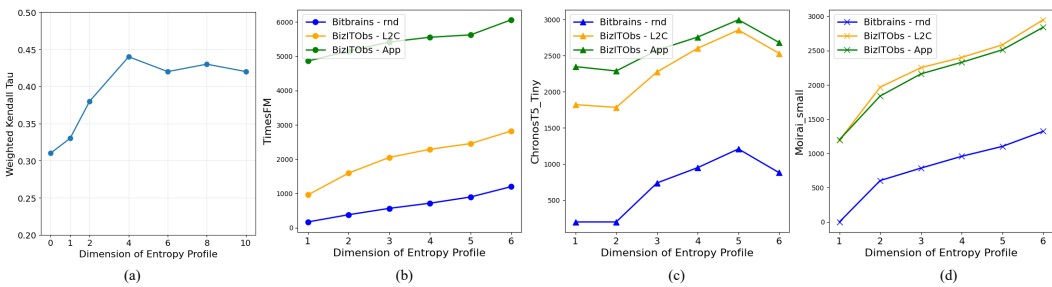

(a)  (b)  (c)  (d)

Figure E: (**a**) Influence of the entropy-profile dimensionality on transferability estimation. (**b–d**) Entropy profiles of models across three datasets with different characteristics.

entropy of middle-layer features exhibits a positive correlation with both finetuned and zero-shot performance. Zheng et al. (2017) suggested that higher entropy indicates a more informative feature space in the pre-trained model, which is not overly biased toward any single pattern and therefore demonstrates greater transferability.

**Dimensionality of the entropy profile.** We also evaluate how the dimensionality of the entropy profile affects transferability estimation, as shown in Figure E (a). When a model has fewer layers than the predefined entropy-profile dimensionality, we pad the profile by repeating the final layer's entropy value. When a model has more layers, we apply average pooling to downsample it to the target length. We observe that removing the entropy profile or using fewer than four dimensions significantly degrades estimation performance. Once the dimensionality reaches four or more, further increases yield no additional gains. Additionally, deeper models typically exhibit smoother entropy evolution across layers; thus, for current TSFMs with up to 32 layers, applying up to a 4× downsampling does not distort the overall information-flow representation.

**Data influence on the entropy profile**. In Figure E (b-d), we compare entropy profiles across three datasets: Bitbrains-rnd (seasonal, 5T sampling), BizITObs-App (seasonal, 10s sampling), and BizITObs-L2c (trend-dominant, 5T sampling). For datasets with the same sampling frequency, L2c exhibits substantially higher information entropy than rnd, suggesting that trend-dominated signals carry more information than purely seasonal ones. When comparing rnd and App, the App dataset shows a higher entropy having a higher sampling frequency, which may indicate that denser temporal sampling captures richer patterns than lower-frequency signals. Although the entropy profile varies with dataset characteristics, the overall entropy-flow pattern—i.e., the shape of the profile—remains similar within a model family. This similarity provides a useful cue for TIMETIC to identify model similarity.

| Characteristics | Complexity |
|---|---|
| Entropy Profile | $O(N \cdot C)$ |
| Fisher Information | $O(N \cdot C + P^2)$ |
| H-score | $O(N \cdot C + N^2 \cdot D)$ |
| Gradient Statistics | $O(N \cdot C + N \cdot B)$ |

Table H: Computational complexity of model characterization methods

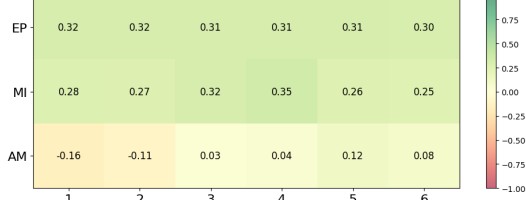

Figure F: Pearson correlation between each dimension of different model (TimesFM_200M) characteristics and finetuned performance.

**Comparison of different model characteristics**. From the perspective of deep model interpretability, many characteristics have been proposed to quantify a model's transferability potential. Examples include Fisher Information (Achille et al., 2019), which measures the sensitivity of model parameters to data; the H-score (Bao et al., 2019b), which captures the distance between the source model's feature or prediction distribution and that of the target task; and Principle Gradient Expectation (Qi et al., 2022), which estimates transferability by comparing gradient differences between the source and target datasets. Although these characteristics may provide effective measures for

transferability, their high computational cost contradicts the core design principle of flexibility in TIMETIC. In contrast, the entropy profile requires only a single forward pass followed by entropy computation, offering a simple yet effective characteristic for transferability estimation. Table H summarizes the computational complexity of these methods, where $N$ is the number of samples for transferability estimation, $C$ is the cost of a TSFM forward pass per sample, $P$ is the total number of TSFM parameters, $D$ is the dimension of the extracted feature vector, and $B$ is the compute cost of a TSFM backward pass.

Under the controlled setting of single-pass inference, we further compare the layer-wise average activation magnitude (AM) and the mutual information (MI) between layer features and input time series with the entropy profile (EP). Following the setup for the entropy profile, we downsample both the average activation magnitude and mutual information sequences to a length of six. As shown in Figure F, the entropy profile and mutual information exhibit similar Pearson correlations with finetuned performance, since feature information entropy essentially represents the upper bound of mutual information. In contrast, the average activation magnitude shows only a weak correlation with finetuned performance.

Table I: Performance comparison of different regressors in TIMETIC. Transferability estimation results are reported for medium-term forecasting using a context table of 1,000 rows.

| Method | Datasets | | | | | | | | | | Mean |
|---|---|---|---|---|---|---|---|---|---|---|---|
| | kdd_cup | bizitobs_l2c | electricity | solar | sz_taxi | jena_weather | ett2 | bitbrains_rnd | bizitobs_app | loop_seattle | |
| Lasso | 0.347 | 0.024 | -0.263 | -0.120 | 0.162 | 0.692 | 0.673 | -0.101 | 0.186 | 0.117 | 0.172 |
| XGB | 0.447 | 0.234 | 0.484 | -0.090 | 0.197 | 0.654 | 0.583 | 0.148 | 0.127 | 0.459 | 0.324 |
| CatBoost | 0.345 | 0.204 | 0.426 | -0.029 | -0.028 | 0.598 | 0.558 | 0.207 | 0.132 | 0.414 | 0.240 |
| FTFormer | 0.227 | -0.321 | 0.092 | -0.148 | 0.058 | 0.525 | 0.420 | 0.005 | 0.140 | -0.448 | 0.055 |
| SAINT | 0.256 | -0.024 | 0.233 | 0.275 | 0.183 | 0.232 | 0.540 | -0.058 | 0.072 | -0.070 | 0.164 |
| TabPFN | 0.137 | 0.522 | 0.426 | 0.574 | 0.061 | 0.561 | 0.536 | 0.530 | 0.485 | 0.459 | 0.429 |

## C.4 DISCUSSIONS ON TABULAR FOUNDATION MODEL

**Comparison of different regressors.** Table I reports the transferability estimation performance of TIMETIC with various regressors, including sparse linear models (Lasso), tree-based models (XGB and CatBoost), and tabular expert models (FTFormer (Gorishniy et al., 2021) and SAINT (Somepalli et al., 2021)). For these methods, we train using a 1,000-row table, whereas TabPFN uses the table directly as context at inference time. TabPFN achieves the best transferability estimation performance, with XGB ranking second. Figure G further compares prediction error and Spearman correlation: although TabPFN's prediction error is comparable to that of other tabular models, it achieves a higher ranking correlation. In contrast, SAINT shows the opposite behavior: despite having a prediction error comparable to other models, it achieves the lowest Spearman correlation. This likely stems from overfitting to the training datasets—its predictions collapse toward the expected value of the target. Although this yields small absolute errors, such predictions fail to preserve ranking information and therefore perform poorly in transferability estimation.

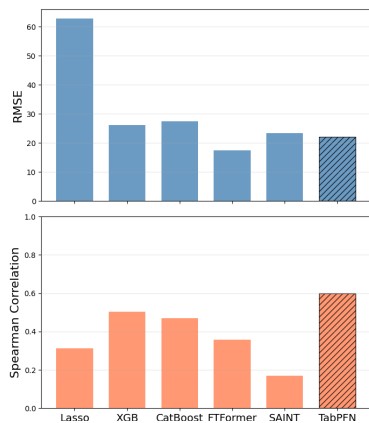

Figure G: Average RMSE (top) and Spearman correlation (bottom) of predictions from different regressors.

**Motivation for introducing TabPFN.** (1) *Strong general regression capability*: TabPFN is trained on a large collection of tabular datasets, giving it a strong ability to model relationships among features. Works (Hollmann et al., 2025; Grinsztajn et al., 2025) have shown that TabPFN achieves superior zero-shot regression performance on multiple large open benchmarks (TabArena, AutoML, and OpenML-CTR23) compared to classical regressors that require training. (2) *Flexible in-context learning capability*: Transferability estimation scenarios naturally produce context tables of varying sizes—ranging from only a few fine-tuning results to continuously growing collections over time.

TabPFN relies on in-context learning, requires no training, and supports a wide range of context table sizes, allowing the table to be reorganized or expanded freely with no retraining cost. (3) *No hyper-parameter tuning*: Standard regressors require careful adjustment of training hyper-parameters for different context table sizes to ensure convergence and avoid overfitting. As a foundation model, TabPFN offers a more robust solution without any hyper-parameter optimization.

### C.5 DISCUSSIONS ON MODEL SELECTION EFFICIENCY

**Enumerative finetuning cost**. Using the Electricity dataset with 12M time points as an example, TimesFM with 500M parameters takes about 15 hours 30 minutes per epoch on a single A100 GPU. And Moirai_small with 14M parameters requires approximately 4 hours 30 minutes to finetune for one epoch. The overall finetuning time cost is therefore substantial, especially it will constantly increases when evaluating a larger candidate model zoo. Moreover, since not all TSFMs share a unified training pipeline, significant human effort is also required to set up training procedures, further increasing the overall cost.

**TimeTic finetuning cost.** In TIMETIC, context construction requires only limited offline finetuning on a few datasets, decoupling the one-time finetuning cost from the potentially unbounded number of future target scenarios. For example, when estimating the model zoo's performance on the Electricity dataset using finetuning results from ETT2—which contains only 1/764 of Electricity's time points—finetuning TimesFM (500M parameters) on ETT2 takes just 12 minutes. Constructing a context table that aggregates the entire model zoo's finetuning experience therefore requires only about 1.3% of the time needed for full enumerative finetuning. With TIMETIC, we can estimate the model ranking on Electricity dataset with only about 1.3% finetuning time.

## D UNCERTAINTY ANALYSIS

We define the performance estimation task as modeling the conditional distribution $p(y|x)$, where $y$ denotes a model's actual fine-tuned performance on the raw time series $x$. The optimal performance of a regressor $f_\theta$ is fundamentally limited by the *aleatoric uncertainty*, $\mathrm{Var}(y|x)$, inherent in the true distribution $p(y|x)$. Formally, the expected squared error of a pointwise regressor $f_\theta$ for each input $x$ is lower-bounded by this variance:

$$\mathbb{E}_{y \sim p(y|x)}\big[(y - f_\theta(x))^2\big] \ \geq \ \mathrm{Var}(y \mid x).$$

In practice, however, observations are restricted to feature-based representations $\phi(x)$, which only partially capture $x$. As a result, the regressor cannot distinguish between states where $\phi(x) = \phi(x')$ but $x \neq x'$. This induces additional *epistemic uncertainty*, raising the lower bound of the expected error from $\mathrm{Var}(y|x)$ to the larger $\mathrm{Var}(y|\phi(x))$:

$$\mathbb{E}_{y \sim p(y|x)}\big[(y - f_\theta(x))^2\big] \ \geq \ \mathrm{Var}(y \mid \phi(x)).$$

Similar bounds also hold for regression-derived metrics such as rank correlations: if multiple $y$-values share identical feature representations $\phi(x)$, their relative rankings cannot be determined. Hence, to minimize epistemic uncertainty, it is crucial for the regressor to incorporate as many informative features as possible. This insight motivates our use of `TotalVariance` as a practical proxy for epistemic uncertainty and explains why TIMETIC emphasizes rich feature and model characterizations to improve transferability estimation. Moreover, `TotalVariance` can also serve as an uncertainty metric to guide context table construction, where minimizing it helps reduce the lower bound of estimation error.

## E USE OF LARGE LANGUAGE MODELS

In preparing this paper, we used large language models solely to improve the clarity and readability of the writing. All substantive research contributions, including conceptualization, model design, experimentation, and analysis, were conducted entirely by the authors.

