# OpenReview forum: "Estimating Time Series Foundation Model Transferability via In-Context Learning"
_ICLR.cc/2026/Conference — Submitted to ICLR 2026_

### Official Review · Reviewer_piGM · 2025-10-26

**Soundness:** 2
**Presentation:** 3
**Contribution:** 1
**Rating:** 2
**Confidence:** 4

**Summary:**

In this paper the authors propose TimeTIC, a framework that estimates the fine-tuning performance ranking of a list of time series foundation models on a specific target dataset. The authors proposed to featurize the dataset by a selection of its statistical characteristics, and to featurize a TSFM by its per-layer entropy profile. A table of dataset features, model features and the FT'ed model's performances is created accordingly and is extrolated by a tabular foundation model onto new models and new datasets. The authors show empirically that the proposed method improves over the ranking list based on zero shot performance.

**Strengths:**

The paper attempts to answer a legit practical question of pretrained base model selection without paying the cost for fine-tuning. Its writing is rather concise and clearly. The proposed entropy profiling of transformer based TSFM is novel.

**Weaknesses:**

Though the research question is legit, the setup proposed in this paper is quite dubious that it's unclear about the actual contribution.
(1) Fine-tuning is still required to construct the context part of the tabular data.
(2) More importantly, all models are fine-tuned under the same setup whereas there is no discussion regarding what is the best fine-tuning setup given a specific pretrained model and a target dataset, which is in fact a more correct and practically useful question to ask here.
(3) It is unclear why a tabular foundation needs to be involved.

In addition, the paper might have ignored some details regarding its models and datasets in the empirical study section which undermines its credibility.

See Questions for details.

**Questions:**

1. In the tabular part, y_context is essentially a function of the pretrain model, the target dataset, and the fine-tuning setup. Can you clarify if the study assumes that the fine-tuning setup is fixed? If so how to justify the conclusions in the paper when a common practice is to optimize the fine-tuning hyperparameters?

2. An alternative of doing time series dataset featurization dimension reduction and applying a pretrained tabular model that I can think of is to NOT do any dimension reduction, and apply a sparse estimator (e.g., the classic LASSO) to estimate the fine-tuned performance. The dimension reduction is basically required by the tabular foundation model as it cannot cope with inputs of very high dimension - why a tabular model along with the in-context notion is introduced here in the first place?

3. For the empirical study, especially regarding the 10 selected datasets, some of them are actually part of the pretraining data of the selected TSFM. For example, Moirai 1.0 and TimesFM 2.0 (the 500m) may both have used the sz-taxi dataset included in LOTSA, Chronos may have used solar included in Chronos data. How does this fact a target dataset being in distribution vs zero shot affect your conclusions in your benchmark?

---

> ### Author Response · Authors · 2025-11-26
> **Response to Reviewer piGM [Part 1]**
>
> We thank the reviewer for the valuable comments and respond to the questions as follows.
>
> > W1: Though the research question is legit, the setup proposed in this paper is quite dubious that it's unclear about the actual contribution. (1) Fine-tuning is still required to construct the context part of the tabular data.
>
> Fine-tuning cost is substantially reduced in TimeTIC.
>
> (1) In $\underline{\text{Appendix C.5 of the revised paper}}$, we add a discussion on fine-tuning cost and model-selection efficiency. **TimeTIC reduces fine-tuning cost by decoupling it from future target scenarios, requiring only limited offline fine-tuning to construct the context.**
>
> For example, consider estimating the model zoo’s performance on the Electricity dataset using finetuning results from ETT2. ETT2 contains only 1/764 as many time points as Electricity. Finetuning TimesFM (500M parameters) on Electricity requires 15 hours and 30 minutes per epoch, whereas finetuning the same model on ETT2 takes only 12 minutes.**Thus, constructing a context table that aggregates the entire model zoo’s finetuning experience requires only about 1.3% of the compute needed for full enumerative finetuning on Electricity.** With TimeTic, we can therefore estimate the model ranking on Electricity using dramatically less finetuning—performed once on a much smaller dataset (ETT2).
>
> (2) As demonstrated in $\underline{\text{ Figure 6: Effect of context table size on transferability estimation performance }}$, **TimeTic continues to provide reliable model-ranking estimates with a context table as small as 100–1000 rows**. This substantially reduces dependence on large-scale offline finetuning, in contrast to other learning-based transferability estimators that require extensive finetuned examples.
>
> > W1.2: More importantly, all models are fine-tuned under the same setup whereas there is no discussion regarding what is the best fine-tuning setup given a specific pretrained model and a target dataset, which is in fact a more correct and practically useful question to ask here.
> >
> > Q1: In the tabular part, y_context is essentially a function of the pretrain model, the target dataset, and the fine-tuning setup. Can you clarify if the study assumes that the fine-tuning setup is fixed? If so how to justify the conclusions in the paper when a common practice is to optimize the fine-tuning hyperparameters?
>
> (1) We adopt a consistent fine-tuning setup across all datasets and TSFMs, ensuring that every model can stably converge. This practice is also commonly used in prior work [1] [2] on transferability estimation.
>
> (2) **Using a consistent fine-tuning setup is a practical trade-off between computational cost and experiment design**. Our chosen setup ensures that all models converge reliably, and additional hyperparameter tuning yields only marginal performance gains. While a full grid search over multiple hyperparameter combinations could potentially provide improvements, it would also incur prohibitively large computational overhead. Such cost is incompatible with our goal of systematically evaluating 10 models across 10 datasets, three forecasting horizons.
>
> (3) **Using a consistent fine-tuning setup does not compromise the validation of TimeTic’s effectiveness.** While hyper-parameter optimization may change the absolute finetuned performance values, it does not alter the underlying functional relationship that TimeTic learns from the context table. The two core capabilities of TimeTic—(a) accurate transferability estimation with arbitrary context-table sizes and (b) generalization to unseen models—are rigorously validated under a unified fine-tuning configuration. These capabilities would remain valid under an optimally tuned setup as well, since hyper-parameter changes affect only the target values, not the mechanisms by which TimeTic leverages entropy profiles and in-context learning.
>
> [1] Pándy M, Agostinelli A, Uijlings J, et al. Transferability estimation using bhattacharyya class separability[C]//Proceedings of the IEEE/CVF Conference on Computer Vision and Pattern Recognition. 2022: 9172-9182.
>
> [2] Nguyen C N, Tran P, Ho L S T, et al. Simple transferability estimation for regression tasks[C]//Uncertainty in Artificial Intelligence. PMLR, 2023: 1510-1521.

---

> ### Author Response · Authors · 2025-11-26
> **Response to Reviewer piGM [Part 2]**
>
> > w1.3: It is unclear why a tabular foundation needs to be involved.
> > Q2: An alternative of doing time series dataset featurization dimension reduction and applying a pretrained tabular model that I can think of is to NOT do any dimension reduction, and apply a sparse estimator (e.g., the classic LASSO) to estimate the fine-tuned performance. The dimension reduction is basically required by the tabular foundation model as it cannot cope with inputs of very high dimension - why a tabular model along with the in-context notion is introduced here in the first place?
>
> Although many regressors are available, we choose to involve in-context learning and a tabular foundation model—TabPFN for several reasons:
>
> 1. **In-context learning for flexible and expandable contexts.** Transferability estimation scenarios naturally produce context tables of varying sizes—ranging from only a few fine-tuning results to continuously growing collections over time. Traditional regressors must be retrained whenever the context changes, which severely limits flexibility. In contrast, TabPFN relies on in-context learning, requires no training, and supports a wide range of context table sizes, allowing the table to be reorganized or expanded freely with no retraining cost.
>
> 2. **No hyper-parameter tuning.** Standard regressors require careful adjustment of training hyper-parameters for different context table sizes to ensure convergence and avoid overfitting. As a foundation model, TabPFN offers a more robust solution without any hyper-parameter optimization.
>
> 3. **Stronger general regression capability.** [1] and [2] have shown that TabPFN’s zero-shot performance—using only in-context learning—already surpasses most classical regressors that require training, including linear models, SVMs, and tree-based methods such as Random Forest, XGBoost, and CatBoost, across multiple large-scale tabular benchmarks (TabArena, AutoML, and OpenML-CTR23).
>
> To further clarify the motivation to choose TabPFN, we compare TabPFN with several regressors—XGBoost, CatBoost, FT-Transformer, SAINT, and LASSO—within the TimeTic framework. The table below reports the weighted Kendall–Tau averaged across 10 datasets under the standard medium-term forecasting setting. TabPFN achieves the highest transferability estimation performance.
>
> | LASSO | XGB | CatBoost | FT-Former | SAINT | TabPFN |
> | ----  | ----  | ----  | ----  | ----  | ----  |
> | 0.172 | 0.324 | 0.240 | 0.055 | 0.164 | 0.429 |
>
> In the $\underline{\text{Appendix C.4 of the revised paper}}$, detailed results—including transferability estimation performance, Spearman correlation, and regression error—are provided.
>
> [1] Hollmann N, Müller S, Purucker L, et al. Accurate predictions on small data with a tabular foundation model[J]. Nature, 2025, 637(8045): 319-326.
>
> [2] Grinsztajn L, Flöge K, Key O, et al. TabPFN-2.5: Advancing the State of the Art in Tabular Foundation Models[J]. arXiv preprint arXiv:2511.08667, 2025.
>
> > W2: The paper might have ignored some details regarding models and datasets which undermines credibility.
>
> Thank you for your feedback.
>
> - Introduction of all models used in our study are provided in $\underline{\text{ Appendix B.1}}$. $\underline{\text{In Table B of the revised Appendix}}$, we additionally include the sources of all pretrained weights to ensure full transparency.
>
> - Dataset descriptions are provided in $\underline{\text{Appendix B.2}}$. To further improve transparency, we additionally list the sources of all datasets in $\underline{\text{Table A of the revised Appendix}}$.

---

> ### Author Response · Authors · 2025-11-26
> **Response to Reviewer piGM [Part 3]**
>
> > Q3: For the empirical study, especially regarding the 10 selected datasets, some of them are actually part of the pretraining data of the selected TSFM. For example, Moirai 1.0 and TimesFM 2.0 (the 500m) may both have used the sz-taxi dataset included in LOTSA, Chronos may have used solar included in Chronos data. How does this fact a target dataset being in distribution vs zero shot affect your conclusions in your benchmark?
>
> Data overlap is a known challenge in time series benchmarks because there is no universally recognized pretraining dataset for TSFMs [1]. **While such overlap can influence zero-shot performance, it has only a minor effect on TimeTIC’s primary objectives and conclusions, which rely on finetuning-based context rather than purely zero-shot behavior.**
>
> 1) TimeTIC aims to predict the ranking of models after fine-tuning. **Even for in-distribution data, fine-tuning provides further performance gains, and the relative ranking of models after fine-tuning is our target.**
> 2)  For our experiments, we select datasests from the standard, publicly used time series benchmarks [1].  The majority of our datasets are not explicitly listed in the major public pre-training data collections. **For the potentially overlapping cases, the overall empirical results (averaging across 10 diverse datasets and across 3 forecasting horizons) remain robust, supporting our conclusions on the generalizability of TimeTIC.**
>
> [1] Aksu T, Woo G, Liu J, et al. Gift-eval: A benchmark for general time series forecasting model evaluation[J]. arXiv preprint arXiv:2410.10393, 2024.

---

### Official Review · Reviewer_THP2 · 2025-10-27

**Soundness:** 3
**Presentation:** 3
**Contribution:** 4
**Rating:** 6
**Confidence:** 5

**Summary:**

Overall, this paper is clearly written, well-motivated, and proposes a sensible formulation with solid time series characterization and convincing empirical results. However, the rationale for dismissing vision-derived metrics and, critically, for choosing TabFMs remains under-argued, and several design details (entropy estimator, profile subsampling) need clarification. With tightened justification and minor fixes, it would be a strong accept.

**Strengths:**

- The paper targets the practical need to pick a TSFM to fine-tune without exhaustively training every candidate. The contrast with enumeration (Fig. 1a) and the ICL framing (Fig. 1b) make the motivation concrete. The introduction also explicitly positions prior categories (statistical metrics vs. meta-learning) and why they fall short for TSFMs.

- Casting transferability estimation as characteristics then performance regression with a tabular context is elegant and flexible. The split between an offline context table and online target inference is simple and scalable.

- The feature extraction/selection pipeline is well-designed: >700 features are reduced using the TotalVariance proxy for epistemic uncertainty, yielding a compact feature representation with empirical justification. Model characterization is also interesting. The entropy-profile idea is architecture-agnostic and easy to compute; the paper shows distinctive patterns across families and sizes (Fig. 3-right). The decision to use at most 10k tokens per layer is pragmatic.

- The evaluation is comprehensive in general. A 10×10×3 benchmark, standardized fine-tuning, and both standard and few-shot regimes are valuable. TimeTic consistently beats baselines in Kendall’s τw and achieves ~0.6 Spearman, incl. generalization to unseen models/datasets (Table 1, Fig. 4–5).

**Weaknesses:**

- The intro says most statistical metrics are from image classification and “depend on strong assumptions about the class structure,” but does not unpack which assumptions make them ill-suited for forecasting (e.g., categorical output distributions, class-conditional feature separability, linear link to labels). This reads as asserted rather than demonstrated.

- The paper notes recent advances in TabFMs and adopts TabPFN, but it stops short of articulating why TabPFN’s inductive biases are appropriate for mapping the concatenated data/model characteristics (plus zero-shot) to fine-tuned performance. Strengthening Sec. 3’s final paragraph with concrete reasoning or evidence (e.g., comparisons to simpler regressors or ablations on TabPFN variants) would better justify the choice.

- Sec. 3.2 uses a “continuous entropy estimator (Kraskov et al.)”, but Kraskov et al. is a mutual-information estimator. If a kNN-style differential entropy estimate is used, the estimator variant, k, bias correction, and dimensionality handling (token-embedding dimension) should be specified, as these strongly affect stability. Currently the description reads generic. Also, profiles are downsampled to length 6 but this convenience choice may blur informative differences for deeper models; a sensitivity analysis or a depth-normalized alignment would help.

- Minor writing/formatting issues. For example, the parenthetical “(Kraskov et al.)” is incomplete, and given the point above, potentially misleading - please cite the exact estimator and year. There are also a few small typos (“Ground Turth” in ToC).

**Questions:**

See the weaknesses.

---

> ### Author Response · Authors · 2025-11-26
> **Response to Reviewer THP2 [Part 1]**
>
> Many thanks to Reviewer THP2 for providing the insightful review and comments.
>
> > W1: The intro says most statistical metrics are from image classification and "depend on strong assumptions about the class structure," but does not unpack which assumptions make them ill-suited for forecasting.
>
>
> The core difficulty of most statistical metrics arises from the fundamental difference between classification (discrete output) and forecasting (continuous regression output). Existing transferability metrics rely on strong assumptions that don't hold in forecasting:
>
> - **Assumption on categorical output:** Metrics like NCE and LEEP operate on logits or predicted class probabilities. In TSFMs, the final output is continuous, making these metrics mathematically undefined or nonsensical without arbitrary discretization.
>
> - **Assumption of feature separability:** Classification metrics assess how separable downstream classes are in the feature space. TSFMs aim to capture temporal dependencies and continuous magnitude shifts, where "separability" for continuous targets is ill-defined.
>
> - **Assumption of one-step prediction**: Most transferability metrics are designed for single-step prediction models and do not account for autoregressive behavior or error accumulation. In contrast, TSFMs typically require iterative multi-step predictions, making these metrics less directly applicable.
>
> In the $\underline{\text{second paragraph of Section 2 in the revised paper}}$, we have added a clear illustration for the limitations of statistical metrics.
>
> > W2: The paper notes recent advances in TabFMs and adopts TabPFN, but it stops short of articulating why TabPFN’s inductive biases are appropriate for mapping the concatenated data/model characteristics (plus zero-shot) to fine-tuned performance. Strengthening Sec. 3’s final paragraph with concrete reasoning or evidence (e.g., comparisons to simpler regressors or ablations on TabPFN variants) would better justify the choice.
>
> (1) Comparison of different regressors.
>
> Thanks for the reviewer's constructive comment, we compare TabPFN with several regressors—XGBoost, CatBoost, FT-Transformer, SAINT, and LASSO—within the TimeTic framework. The table below reports the weighted Kendall–Tau averaged across 10 datasets under the standard medium-term forecasting setting. TabPFN achieves the best transferability estimation performance.
>
> | LASSO | XGB | CatBoost | FT-Former | SAINT | TabPFN |
> | ----  | ----  | ----  | ----  | ----  | ----  |
> | 0.172 | 0.324 | 0.240 | 0.055 | 0.164 | 0.429 |
>
> In the $\underline{\text{Appendix C.4 of the revised paper}}$, detailed results—including transferability estimation performance, Spearman correlation, and regression error—are provided.
>
> (2) Inducive bias of TabPFN for transferability estimation.
>
> TabPFN is trained on a large collection of tabular datasets, giving it a strong ability to model relationships among features. Works [1] [2] have shown that **TabPFN achieves superior zero-shot regression performance on multiple large open benchmarks (TabArena, AutoML, and OpenML-CTR23) compared to classical regressors that require training.**
>
> Transferability estimation—predicting fine-tuned performance from data and model characteristics with limited fine-tuning history—naturally fits this zero-shot in-context learning scenario, making TabPFN well-suited for the task.
>
> [1] Hollmann N, Müller S, Purucker L, et al. Accurate predictions on small data with a tabular foundation model[J]. Nature, 2025, 637(8045): 319-326.
>
> [2] Grinsztajn L, Flöge K, Key O, et al. TabPFN-2.5: Advancing the State of the Art in Tabular Foundation Models[J]. arXiv preprint arXiv:2511.08667, 2025.
>
> In $\underline{\text{Appendix C.4 of the revised paper}}$, we further elaborate on the reason for using TabPFN as the regressor within TimeTIC.

---

> ### Author Response · Authors · 2025-11-26
> **Response to Reviewer THP2 [Part 2]**
>
> > W3: Sec. 3.2 uses a “continuous entropy estimator (Kraskov et al.)”, but Kraskov et al. is a mutual-information estimator. If a kNN-style differential entropy estimate is used, the estimator variant, k, bias correction, and dimensionality handling (token-embedding dimension) should be specified, as these strongly affect stability. Currently the description reads generic. Also, profiles are downsampled to length 6 but this convenience choice may blur informative differences for deeper models; a sensitivity analysis or a depth-normalized alignment would help.
>
> - We have corrected the reference to cite the original work [1].
>
> - The KL entropy estimator only has a key hyperparameter, the nearest-neighbor count k. When k=1, it suffers from extremely high variance and instability—especially with finite or high-dimensional data—often collapsing when duplicate points yield zero nearest-neighbor distance. Using a slightly larger value such as k=3, 6, 10 will introduce a small bias but greatly reduces variance, providing a much more stable and reliable entropy estimate. In our experiments, we adopt a blanaced choice of k=6.
>
> - Following the reviewer’s suggestion, we analyze the effect of entropy profile dimensionality in $\underline{\text{Figure E (a) of Appendix C.3 in the revised paper}}$. We find that **removing the entropy profile or using fewer than four dimensions significantly degrades estimation performance. Once the dimensionality reaches four or more, further increases yield no additional gains.**
>
> In $\underline{\text{Section 3.2 of the revised version}}$, we have clarified the computation of entropy profiles and provided further details to improve transparency.
>
>
> [1] Kozachenko L. Sample estimate of the entropy of a random vector[J]. Probl. Pered. Inform., 1987, 23: 9.
>
>
> > W4: Minor writing/formatting issues. For example, the parenthetical “(Kraskov et al.)” is incomplete, and given the point above, potentially misleading - please cite the exact estimator and year. There are also a few small typos (“Ground Turth” in ToC).
>
> We appreciate the reviewer pointing out these minor errors.
> - The incomplete parenthetical citation (Kraskov et al.) has been corrected to specify the exact Kraskov-Stögbauer-Grassberger (KSG) estimator and the relevant publication year, (Kraskov et al., 2004), as requested.
> - The typo "Ground Turth" in the Table of Contents (ToC) has been corrected to "Ground Truth".

---

### Official Review · Reviewer_i572 · 2025-10-31

**Soundness:** 2
**Presentation:** 3
**Contribution:** 1
**Rating:** 4
**Confidence:** 4

**Summary:**

This paper proposes TIMETIC, a practical framework for estimating the fine-tuned performance of Time Series Foundation Models (TSFMs) on unseen datasets without performing actual fine-tuning. The core idea formulates transferability estimation as an in-context prediction task, leveraging a tabular foundation model (TabPFN) to learn the mapping from dataset and model characteristics to expected performance. However, three core components of this paper, including time series characterization, model characterization, and in-context transferability estimation, are merely combinations of existing techniques, lacking sufficient originality or innovation.

**Strengths:**

1. As TSFMs gain increasing attention, efficiently assessing their applicability to downstream tasks without the cost of full fine-tuning has become a critical challenge. TIMETIC directly addresses this bottleneck, offering substantial value for automated model selection, especially in resource-constrained or time-sensitive scenarios.

2. The core idea reframes transferability estimation as an in-context prediction problem, where a tabular foundation model (TabPFN) is prompted to learn the mapping from the characteristics of datasets and models to their expected performance.

3. The introduction of TotalVariance as a proxy for epistemic uncertainty provides a principled basis for feature selection, improving generalization by focusing on the most informative and stable features.

4. Representing models through information entropy profiles across network layers provides a meaningful and interpretable way to encode architectural differences (e.g., encoder-only vs. encoder-decoder).

**Weaknesses:**

1. While the paper formulates transferability estimation as an in-context prediction task, the three core components, time series characterization (using TotalVariance for feature selection), model characterization (based on an entropy estimator), and in-context transferability estimation (relying on the pre-existing TabPFN), are largely integrations of established techniques, offering limited originality or conceptual innovation.

2. I am also puzzled by the computation process of the entropy profile, particularly the step of subsampling each entropy profile to a fixed length of six to ensure consistent representation across models with varying depths. Why is the minimum depth set to six? Given that TSFMs typically have many more layers, how exactly is the subsampling performed across the layers? Could this subsampling cause deeper models to lose important information? And how would the method handle models with depths fewer than the preset value?

3. Although you frame model selection as an in-context learning problem and use TabPFN to predict model performance on unseen datasets, this essentially amounts to a regression task. Why do other standard regressors perform poorly on this task? If the superior performance is largely attributed to the inherent strength of TabPFN, it raises questions about the extent of the proposed method's own contribution.

4. The ground truth fine-tuned performance of various TSFMs in Table C also confuses me. (1) Why do the results of different models differ so significantly on the same dataset? For example, in the short-term forecasting tasks on the solar:H dataset, why is the performance of TimesFM-500M as high as 90.266, while other models are around 1–2? Generally, larger models within the same architecture are expected to perform better. Similar issues appear in medium-term and long-term tasks. (2) Why do the results for some datasets show that short-term forecasting performs much worse than medium-term or long-term, despite long-term forecasting typically being a more challenging task? For instance, on the bizitobs l2c:5T dataset, the short-term result of Chronos-tiny is 22.862, while the long-term result is only 1.140. I believe inaccuracies in these ground truth fine-tuned performances could significantly affect the evaluation of your method.

**Questions:**

1. Please further clarify the originality and conceptual contribution of your method, as the core components are largely based on existing techniques, which currently limits the perceived novelty of the work.

2. Please provide a detailed explanation of the entropy profile computation process, including the rationale for setting the fixed length to six, the specific subsampling strategy used across layers, how models with fewer than six layers are handled, and an analysis of potential information loss in deeper models.

3. Please justify the choice of TabPFN over standard regression models by providing comparative results with alternative regressors. If TabPFN’s strong performance is central to your method, please clearly delineate how much of the success stems from your framework design versus the underlying model’s capabilities.

4. Please verify and explain the ground truth fine-tuned performance values reported in Table C. Specifically, address the extreme performance discrepancies across models on the same dataset (e.g., TimesFM-500M vs. others on solar:H) and the counterintuitive results where short- or medium-term forecasting performs significantly worse than long-term (e.g., on bizitobs l2c:5T), as inaccuracies here could undermine the validity of your evaluation.

---

> ### Author Response · Authors · 2025-11-26
> **Response to Reviewer i572 [Part 1]**
>
> We would like to sincerely thank Reviewer i572 for providing the valuable feedback.
>
> > W1: While the paper formulates transferability estimation as an in-context prediction task, the three core components, time series characterization (using TotalVariance for feature selection), model characterization (based on an entropy estimator), and in-context transferability estimation (relying on the pre-existing TabPFN), are largely integrations of established techniques, offering limited originality or conceptual innovation.
>
> > Q1: Please further clarify the originality and conceptual contribution of your method, as the core components are largely based on existing techniques, which currently limits the perceived novelty of the work.
>
> (1) TimeTic is not the combination of existing techiques.
>
> Our framework is inspired by concepts from model interpretability (entropy profiles) and tabular foundation models (TabPFN), but **it leverages them in a way that has not been explored in prior work.**
>
> First, we use the feature-wise information entropy profile as a model representation, whereas prior work has used it only for interpreting internal model behavior or illustrating generalization properties.
>
> Second, we exploit TabPFN’s in-context learning capability to enable flexible and practical transferability estimation, while TabPFN was previously applied solely to tabular regression tasks.
>
> (2) TimeTic is clearly distinct from existing transferability estimation methods.
>
> First, **TimeTic formulates transferability estimation in a more practical setting—predicting the fine-tuned performance of arbitrary models using any available finetuning history**. In contrast, existing learning-based methods assume a fixed and predefined set of finetuning results.
>
> Second, **TimeTic can incorporate an arbitrary number of finetuning records (from 100 to over 6000) by leveraging TabPFN’s in-context learning capability, as demonstrated in $\underline{\text{Figure 6 (iii)}}$**. Existing methods, however, require numerical finetuning histories to train their estimators and do not naturally scale to large or variable-sized histories.
>
> Third, **TimeTic enables transferability estimation for unseen models through a unified entropy-profile–based model representation, as demonstrated in $\underline{\text{Figure 5}}$**. Existing methods are typically restricted to a fixed model zoo or require additional training to accommodate new models.
>
> (3) The contribution of the transferability estimation benchmark should not be overlooked.
>
> In addition to proposing TimeTic, we introduce a benchmark and codebase spanning 10 models, 10 datasets, three forecasting horizons, and two sample regimes. The construction details are provided in $\underline{\text{Appendix B}}$.
>
> Thus, TimeTic is far beyond simple integration of existing techniques. To highlight this point, we have rephrased the listed contributions in the $\underline{\text{ Introduction of revised paper}}$.

---

> ### Author Response · Authors · 2025-11-26
> **Response to Reviewer i572 [Part 2]**
>
> > W2: I am also puzzled by the computation process of the entropy profile, particularly the step of subsampling each entropy profile to a fixed length of six to ensure consistent representation across models with varying depths. Why is the minimum depth set to six? Given that TSFMs typically have many more layers, how exactly is the subsampling performed across the layers? Could this subsampling cause deeper models to lose important information? And how would the method handle models with depths fewer than the preset value?
>
> > Q2: Please provide a detailed explanation of the entropy profile computation process, including the rationale for setting the fixed length to six, the specific subsampling strategy used across layers, how models with fewer than six layers are handled, and an analysis of potential information loss in deeper models.
>
> (1) The entropy profile computation involves four steps:
>
> 1) Sample $n$ time windows $\lbrace \omega_i \rbrace_{i=1}^n$.
>
> 2) Extract the token embeddings from each model layer $j$ for these windows, denoted $\boldsymbol{h}_j^i$ for window $i$.
>
> 3) Compute the information entropy of the embeddings  $\lbrace \boldsymbol{h}_j^i \rbrace$, $i = 1...n$ at each layer, using at most 10,000 tokens per layer to reduce computational cost, yielding an entropy profile $\boldsymbol{h} \in \mathbb{R}^N$, $N$ is the number of model layer.
> 4) Standardize the entropy profile to a fixed length of six for comparison across models by dividing it into six consecutive segments and averaging within each segment (average pooling).
>
> (2) Implementaion details of entropy profile computation:
>
> 1) We set the length of entropy profile to 6 because the minimum layer of the model in our model zoo is 6.
>
> 2) For models with layers more than the predefined length, we apply an averaging pooling strategy to compress the trajectory while preserving the overall information flow.
>
> 3) For models with layers fewer than the predefined length, we pad the profile by repeating the final layer’s entropy value.
>
> In $\underline{\text{Section 3.2 of the revised version}}$, we have clarified the computation of entropy profiles and provided further details to improve transparency.
>
> (3) The influence of entropy profile downsampling on transferability esimation:
>
> Following the reviewer’s suggestion, we analyze the effect of entropy profile dimensionality in $\underline{\text{Figure E (a) of Appendix C.3 in the revised paper}}$. We find that **removing the entropy profile or using fewer than four dimensions significantly degrades estimation performance. Once the dimensionality reaches four or more, further increases yield no additional gains.**
>
> Deeper models generally exhibit smoother entropy evolution across layers. For our deepest model with 32 layers, downsampling the entropy profile to length six (a 4× reduction) preserves the overall information-flow representation without distortion

---

> ### Author Response · Authors · 2025-11-26
> **Response to Reviewer i572 [Part 3]**
>
> > W3: Although you frame model selection as an in-context learning problem and use TabPFN to predict model performance on unseen datasets, this essentially amounts to a regression task. Why do other standard regressors perform poorly on this task? If the superior performance is largely attributed to the inherent strength of TabPFN, it raises questions about the extent of the proposed method's own contribution.
>
> > Q3: Please justify the choice of TabPFN over standard regression models by providing comparative results with alternative regressors. If TabPFN’s strong performance is central to your method, please clearly delineate how much of the success stems from your framework design versus the underlying model’s capabilities.
>
> (1) Comparison of different regressors.
>
> As requested by the reviewer, we compare TabPFN with several regressors—XGBoost, CatBoost, FT-Transformer, SAINT, and LASSO—within the TimeTic framework. The table below reports the weighted Kendall–Tau averaged across 10 datasets under the standard medium-term forecasting setting. TabPFN achieves the highest transferability estimation performance.
>
> | LASSO | XGB | CatBoost | FT-Former | SAINT | TabPFN |
> | ----  | ----  | ----  | ----  | ----  | ----  |
> | 0.172 | 0.324 | 0.240 | 0.055 | 0.164 | 0.429 |
>
> In the $\underline{\text{Appendix C.4 of the revised paper}}$, detailed results—including transferability estimation performance, Spearman correlation, and regression error—are provided.
>
> (2) Weakness of traditional regressors.
>
> $\underline{\text{Figure G in Appendix C.4 of the revised paper}}$ reports the prediction error of different regressors. Except for LASSO, the other four regressors achieve similar low errors; however, **these predictions likely result from overfitting to the training data, collapsing toward the target’s mean. While this reduces absolute error, it fails to preserve ranking information, leading to relatively poor performance in transferability estimation**.
>
> **Traditional regressors typically require careful hyperparameter tuning to mitigate overfitting, whereas TabPFN provides robust results without any hyperparameter optimization.**
>
> (3) TabPFN’s in-context learning capability is important for TimeTIC’s flexibility, but it represents only one part of the framework’s core contributions. TimeTIC integrates multiple key components:
>
> - Data characterization: Selecting a concise set of representative features from a large candidate pool to avoid the curse of dimensionality and ensure regressor stability, as illustrated in $\underline{\text{Figure 3 (left)}}$.
> - Model characterization: Introducing the entropy profile, an inference-time descriptor that enables generalization across unseen models, supported by $\underline{\text{Figure 5 and Figure D of Appendix C.3}}$.
> - In-context transferability estimation: Predicting fine-tuned performance of arbitrary models using any available fine-tuning history, framing transferability estimation in a practical and flexible manner.
>
> **TabPFN acts as the in-context regressor that leverages these components, but the novelty lies in the overall framework and the synergy among its parts.**

---

> ### Author Response · Authors · 2025-11-26
> **Response to Reviewer i572 [Part 4]**
>
> > W4: The ground truth fine-tuned performance of various TSFMs in Table C also confuses me. (1) Why do the results of different models differ so significantly on the same dataset? For example, in the short-term forecasting tasks on the solar:H dataset, why is the performance of TimesFM-500M as high as 90.266, while other models are around 1–2? Generally, larger models within the same architecture are expected to perform better. Similar issues appear in medium-term and long-term tasks. (2) Why do the results for some datasets show that short-term forecasting performs much worse than medium-term or long-term, despite long-term forecasting typically being a more challenging task? For instance, on the bizitobs l2c:5T dataset, the short-term result of Chronos-tiny is 22.862, while the long-term result is only 1.140. I believe inaccuracies in these ground truth fine-tuned performances could significantly affect the evaluation of your method.
>
> > Q4: Please verify and explain the ground truth fine-tuned performance values reported in Table C. Specifically, address the extreme performance discrepancies across models on the same dataset (e.g., TimesFM-500M vs. others on solar:H) and the counterintuitive results where short- or medium-term forecasting performs significantly worse than long-term (e.g., on bizitobs l2c:5T), as inaccuracies here could undermine the validity of your evaluation.
>
> Many thanks to the reviewers for the careful feedback. We re-examined the fine-tuning performance collection pipeline and have corrected the reported issues.
>
> (1) TimesFM anomaly on solar:H
>
> The unusually high TimesFM-500M value resulted from an error during the CSV-to-LaTeX table conversion. This has been corrected in the revised version.
>
> (2) Discrepancies between short-, medium-, and long-term results. The unexpected discrepancies arise from three factors:
>
> - Seasonal period issue (main reason): The unusually large short-term MASE on bizitobs l2c:5T, bizitobs application and LOOP_SEATTLE was primarily caused by not using a consistent seasonal period as medium/long-term when computing MASE. These datasets have a sampling frequency of 5min and a seasonal period of 288. However, for short-term forecasting, our code set the seasonal period to 1, resulting in inconsistent normalization and inflated short-term MASE values. **Importantly, this does not affect the transferability-estimation evaluation, as the same seasonal period is applied to all models, preserving performance rankings.**
> - Different test samples across forecasting settings: We use sliding windows with strides equal to the forecast horizon. Because horizons differ across short-, medium-, and long-term forecasting, the resulting test windows also differ. Thus, direct numerical comparison across forecasting settings is not appropriate.
> - Model-dependent horizon preference: Some models inherently perform better on medium- or long-term tasks than on short-term ones, a pattern also observed in prior benchmarks [1][2].
>
> We have correcred the above issues and updated the $\underline{\text{Table C in the revised manuscript}}$.
>
> [1] Wu H, Xu J, Wang J, et al. Autoformer: Decomposition transformers with auto-correlation for long-term series forecasting[J]. Advances in neural information processing systems, 2021, 34: 22419-22430.
>
> [2] Aksu T, Woo G, Liu J, et al. Gift-eval: A benchmark for general time series forecasting model evaluation[J]. arXiv preprint arXiv:2410.10393, 2024.

---

### Official Review · Reviewer_4Tz1 · 2025-10-31

**Soundness:** 3
**Presentation:** 3
**Contribution:** 2
**Rating:** 4
**Confidence:** 3

**Summary:**

This paper tackles how to quickly choose the best TSFM to fine-tune under limited downstream data. The authors recast model transferability estimation as a table-like ICL problem: they organize mappings of (target dataset features + candidate model representations + historical (dataset, model) → post-fine-tuning performance) into a context table, and use a pretrained tabular model (TabPFN) to predict performance without updating parameters. They also propose a layer-wise entropy profile as an architecture-agnostic model representation.

**Strengths:**

The pipeline is easy to follow: dataset feature extraction → model representation → table construction → ICL inference.

The evaluation covers multiple domains, frequencies, and horizons, and tests three generalization settings (unseen datasets, unseen models, and both unseen).

Nice code is provided, improving reproducibility and practical usability.

**Weaknesses:**

Learner dependence: The method mainly relies on TabPFN. Although its ICL nature fits the “no retraining” goal, there is no comparison with other tabular learners (e.g., FT-Transformer, SAINT, XGBoost/CatBoost).

Model representation evidence: The entropy profile is promising, but evidence is limited on why it reliably reflects “fine-tune-ability” across TSFMs. Suggested additions:

Systematic comparisons against alternative representations (feature entropy/information gain, Fisher information, H-score variants, NCE, gradient/activation statistics).

Robustness under shift (sampling rate changes, seasonality, spikes): does the profile spuriously vary with frequency or scaling?

Motivation and cost trade-off:

Since TSFMs are small and cheap to fine-tune, is predicting post-fine-tuning performance a strong practical need? How many time can you save with different GPUs.

The method requires offline computation across multiple (dataset, model) pairs to build the context, which may conflict with the “reduce cost” goal.
Consider the cost of finetuning, I would rather simple choose ensembling a stronger and often more stable baselinet

**Questions:**

Please check weakness

---

> ### Author Response · Authors · 2025-11-26
> **Response to Reviewer 4Tz1 [Part 1]**
>
> We would like to sincerely thank Reviewer R4TZ1 for providing a detailed review and insightful suggestions.
>
> > W1: Learner Dependence. The method mainly relies on TabPFN. Although its ICL nature fits the "no retraining" goal, there is no comparison with other tabular learners (e.g., FT-Transformer, SAINT, XGBoost/CatBoost).
>
> As requested by the reviewer, we compare TabPFN with several regressors—XGBoost, CatBoost, FT-Transformer, SAINT, and LASSO—within the TimeTic framework. The table below reports the weighted Kendall–Tau averaged across 10 datasets under the standard medium-term forecasting setting. TabPFN achieves the highest transferability estimation performance.
>
> | LASSO | XGB | CatBoost | FT-Former | SAINT | TabPFN |
> | ----  | ----  | ----  | ----  | ----  | ----  |
> | 0.172 | 0.324 | 0.240 | 0.055 | 0.164 | 0.429 |
>
> In the $\underline{\text{ Appendix C.4 of the revised paper}}$, detailed results—including transferability estimation performance, Spearman correlation, and regression error—are provided.
>
>
> > W2: Model Representation Evidence:
> > 1. The entropy profile is promising, but evidence is limited on why it reliably reflects "fine-tune-ability" across TSFMs. Suggested additions:
> > 2. Systematic comparisons against alternative representations (feature entropy/information gain, Fisher information, H-score variants, NCE, gradient/activation statistics).
> > 3. Robustness under shift (sampling rate changes, seasonality, spikes): does the profile spuriously vary with frequency or scaling?
>
> 1. In the $\underline{\text{ Figure D of Appendix C.3 in the revised paper }} $ we visualize the Pearson correlation between each entropy-profile dimension and the finetuned performance. **We observe that the information entropy of middle-layer features maintains a positive correlation of around 0.1~0.3**. A possible explanation, following [1], is that higher entropy reflects a more informative and less pattern-biased feature space, which in turn suggests stronger transferability.
>
> [1] Zheng G, Sang J, Xu C. Understanding deep learning generalization by maximum entropy[J]. arXiv preprint arXiv:1711.07758, 2017.
>
> 2. In the $\underline{\text{ Last Paragraph of Appendix C.3 in the revised paper}}$, we discuss and compare other potential model characterization methods.
>
> - **For Fisher Information, H-score variants, and Gradient Statistics, the methods will bring substantial extra compute budget, which contradicts the core design principle of flexibility in TimeTic**. Entropy profile requires only a single forward pass followed by entropy calculation, offering a simple yet effective characteristic for transferability estimation. The compute complexity of these methods are shown below as well as in  $\underline{\text{ Table H}}$, where $N$ is the number of samples for transferability estimation, $C$ is the cost of a TSFM forward pass per sample, $P$ is the total number of TSFM parameters, $D$ is the dimension of the extracted feature vector, and $B$ is the compute cost of a TSFM backward pass.
>
> | Characteristics | Entropy Profile | Fisher Information | H-score | Gradient Statistics |
> | ------- | ------- | ------- | ------- | ------- |
> | Complexity | $O(N⋅C)$ | $O(N⋅C+P^2)$ | $O(N⋅C+N^2⋅D)$ | $O(N⋅C+N⋅B)$ |
>
> - Under a controlled single-pass inference setting, we compare layer-wise average activation magnitude and the mutual information between layer features and the input time series against the entropy profile. In  $\underline{\text{ Figure F }}$, **entropy profile and mutual information show similar Pearson correlations with finetuned performance**, as feature entropy is essentially the upper bound of mutual information. In contrast, **average activation magnitude exhibits only a weaker correlation with finetuned performance**.
>
> 3. In  $\underline{\text{ Figure E of Appendix C.3}}$, we analyze how data characteristics affect the entropy profile. Our findings show that the **variation is meaningful rather than spurious—it reflects properties of the input data and the model’s response to them**:
>
> - Sampling frequency: Higher sampling frequency leads to higher information entropy, suggesting denser sampling captures richer temporal patterns.
>
> - Signal characteristics: Trend-dominated datasets exhibit higher entropy than seasonal ones, indicating that repeated season cannot increase the information amount of features.
>
> - Model-family consistency: Although absolute entropy values shift with the dataset, the shape of the entropy profile—the entropy-flow pattern—remains consistent within a model family. This stable pattern provides a reliable cue for TimeTic to identify model similarity.

---

> ### Author Response · Authors · 2025-11-26
> **Response to Reviewer 4Tz1 [Part 2]**
>
> > W3: Motivation and Cost Trade-off.
> > 1. Since TSFMs are small and cheap to fine-tune, is predicting post-fine-tuning performance a strong practical need? Consider the cost of finetuning, I would rather simple choose ensembling a stronger and often more stable baseline.
> > 2. How many time can you save with different GPUs. The method requires offline computation across multiple (dataset, model) pairs to build the context, which may conflict with the "reduce cost" goal.
>
> In $\underline{\text{Appendix C.5 of the revised paper}}$, we further discuss finetuning cost and model selection efficiency.
>
> - **Enumerative finetuning remains costly, and the cost grows with the size of the model zoo**. Using the Electricity dataset with 12M time points as an example, TimesFM (500M parameters) requires roughly 15 h 30 min per epoch on a single A100 GPU, and Moirai_small (14M parameters) still takes 4 h 30 min per epoch. Moreover, because TSFMs lack a unified training pipeline, substantial human effort is required to configure training for each model, further increasing the overall cost.
>
> - **TimeTic decouples finetuning cost from future target scenarios by requiring only limited offline finetuning for context construction.** For instance, when estimating the model zoo’s performance on Electricity using finetuning results from ETT2—which contains only 1/764 of Electricity’s time points—finetuning TimesFM (500M) on ETT2 takes only 12 minutes. **Constructing a context table that aggregates the entire model zoo’s finetuning experience thus requires only ≈1.3% of the time needed for full enumerative finetuning.**
> With TimeTic, we can estimate the model ranking on Electricity using dramatically less finetuning, performed once on a much smaller dataset (ETT2).

---

### Author Response · Authors · 2025-11-27
**Response to all reviewers & Summary of revisions**

We sincerely thank all the reviewers for their insightful comments and valuable suggestions, which have greatly helped us improve our paper.

This paper introduces TimeTIC, the first in-context transferability estimation framework. It leverages **unified tabular data/model representations** and unlocks the **in-context learning capabilities** of tabular foundation models to enable more flexible and practical transferability estimation. Across **experiments on 10 datasets, 10 models, and 3 forecasting settings**, we demonstrate that TimeTIC outperforms existing baselines and generalizes well to predicting the transferability of unseen models.

The reviewers expressed positive views on our work, describing the addressed problem as a “**legit practical question**”; the method as “pipeline is **easy to follow**”, “offers **substantial value** for automated model selection”, “notes the **recent advancement** in tabular foundation models”, and "the proposed entropy profiling of Transformer-based TSFMs as **novel**”. They also commented that the experiments include “**nice code is provided**” and that the “**evaluation is comprehensive in general.**”

The reviewers also raised insightful and constructive concerns. We made every effort to address these concerns by providing detailed clarification and requested results. Here is the summary of the major revisions:

1) **Illustrate the motivation for introducing TabPFN (Reviewer 4Tz1, i572, THP2, and piGM)**:
 - We added new discussions in $\underline{\text{Appendix C.4}}$ explaining why TabPFN is adopted, highlighting three main reasons:
(1) strong general regression performance;
(2) flexible in-context learning capability;
(3) No hyper-parameter tuning.
 - Also we added 3 groups of experimental results comparing TabPFN with five classic regressors in terms of transferability estimation accuracy, prediction error, and Spearman correlation, where TabPFN shows an overall superior performance.

2) **Elaborate the implementation details and behavior of entropy profile (Reviewer 4Tz1,  i572, THP2)**:
- We clarified the computation procedure and hyperparameter choices for the entropy profile (EP) in $\underline{\text{Section 3.2}}$.
- Additionally, we added four experiments in $\underline{\text{Appendix C.3}}$ to further illustrate: (1) EP’s positive correlation with fine-tuned performance, (2) that an EP dimensionality ≥ 4 is sufficient for transferability estimation, (3) EP’s reasonable adaptation to changes in data properties or frequency, and (4) that EP has lower computational complexity and relatively strong correlation with fine-tuned performance compared with other model characteristics.

3) **Discuss the effiency of offline finetuning (Reviewer 4Tz1, i572 and piGM)**:
- We illustrate that TimeTIC can estimate transferability with minimal offline cost using only a small number of finetuning results (100–1,000), as shown in $\underline{\text{Figure 6(iii)}}$.
- Additionally, we present a case of selecting models on the large Electricity dataset by leveraging finetuning results from the smaller ETT2 dataset, where TimeTIC requires only about 1.3% of the time compared with exhaustive model selection. Further details are provided in $\underline{\text{Appendix C.5}}$.

4) **Explain the settings and results of benchmark(Reviewer i572, and piGM):**
- We further improved benchmark transparency by including dataset sources and model weight sources in $\underline{\text{Appendix B.1 and B.2}}$.
- Additionally, we re-examined our benchmark construction pipeline and corrected errors in the reported results.

We have added **7 new groups of experimental results** to address the above concerns. The revised content and sections are $\underline{\text{highlighted in blue in the revised manuscript}}$.

The reviewers’ valuable suggestions have been extremely helpful in improving our paper. We are happy to answer any further questions.

We look forward to your feedback.

---

### Meta-Review · Area_Chair_mjJc · 2026-01-06

**Summary:**

This paper studies an important and timely problem: estimating the transferability of time series foundation models (TSFMs) in order to guide downstream fine-tuning. The proposed framework, TimeTIC, formulates transferability estimation as an in-context learning problem over tabular representations that combine dataset characteristics, model characteristics, and historical fine-tuning results. Reviewers generally agreed that the paper is clearly written, technically competent, and addresses a legitimate practical question. The benchmark construction and the use of entropy-based model characterization were also viewed as interesting.

However, the central issue is that, despite substantial engineering effort and extensive experimentation, the paper’s conceptual contribution and practical impact remain limited relative to its complexity. Multiple reviewers raised concerns that the core components—dataset featurization, entropy-profile-based model representation, and tabular in-context regression—are largely integrations of existing ideas rather than a clear methodological advance. While the authors argue that the novelty lies in the overall framework and synergy among components, this claim remains insufficiently convincing.

Moreover, the practical motivation is not fully compelling. Although TimeTIC aims to reduce fine-tuning cost, the method still relies on non-trivial offline fine-tuning to construct the context table, and assumes a unified fine-tuning setup across models and datasets. Reviewers questioned whether predicting fine-tuned performance under such constraints meaningfully reflects real-world model selection practice, where fine-tuning strategies are typically optimized per model–dataset pair. As a result, it remains unclear whether TimeTIC addresses the most practically relevant version of the transferability problem.

The rebuttal is thorough and addresses many technical questions, adds experiments, and corrects reported issues. Nevertheless, these additions primarily improve clarity and completeness rather than resolving the fundamental concerns about originality, necessity, and real-world impact. Overall, while this is a solid and careful piece of work, its contribution falls short of the level of conceptual clarity and significance expected at ICLR.

**Reviewer Concerns:**

The authors provided a detailed rebuttal with additional experiments and clarifications. However, based on the Area Chair’s assessment, several key concerns remain insufficiently resolved:

Limited conceptual novelty beyond combining existing techniques.

Unclear practical benefit given the continued reliance on offline fine-tuning and a fixed fine-tuning setup.

Heavy dependence on TabPFN, raising questions about how much of the performance stems from the framework versus the underlying regressor.

Methodological complexity that is not clearly justified by decisive empirical gains.

**Reviewer Scores:**

Based on the original reviews and the rebuttal, it is unlikely that reviewer scores would change substantially. Reviewers expressing concerns about novelty and practical relevance would likely maintain their original assessments, while more positive reviewers did not raise arguments strong enough to outweigh these issues.

---

### Decision · Program_Chairs · 2026-01-26

Reject